# Genetic drug target validation using Mendelian randomisation

Amand F. Schmidt [1,2,3,8 ✉], Chris Finan [1,2,8], Maria Gordillo-Marañón [1], Folkert W. Asselbergs[1,3,4], Daniel F. Freitag [5], Riyaz S. Patel[1,2], Benoît Tyl [6], Sandesh Chopade[1,2], Rupert Faraway[1,2,7], Magdalena Zwierzyna [1,2] & Aroon D. Hingorani [1,2,4]

Mendelian randomisation (MR) analysis is an important tool to elucidate the causal relevance of environmental and biological risk factors for disease. However, causal inference is undermined if genetic variants used to instrument a risk factor also influence alternative disease-pathways (horizontal pleiotropy). Here we report how the 'no horizontal pleiotropy assumption' is strengthened when proteins are the risk factors of interest. Proteins are typically the proximal effectors of biological processes encoded in the genome. Moreover, proteins are the targets of most medicines, so MR studies of drug targets are becoming a fundamental tool in drug development. To enable such studies, we introduce a mathematical framework that contrasts MR analysis of proteins with that of risk factors located more distally in the causal chain from gene to disease. We illustrate key model decisions and introduce an analytical framework for maximising power and evaluating the robustness of analyses.

[1] Institute of Cardiovascular Science, Faculty of Population Health, University College London, London WC1E 6BT, UK. [2] UCL BHF Research Accelerator Centre, London, UK. [3] Department of Cardiology, Division Heart and Lungs, University Medical Center Utrecht, Heidelberglaan 100, 3584 CX Utrecht, The Netherlands. [4] Health Data Research UK, 222 Euston Road, London, UK. [5] Bayer AG Pharmaceuticals, Open Innovation & Digital Technologies, Aprather Weg 18a, Wuppertal 42096, Germany. [6] Center for Therapeutic Innovation, Cardiovascular and Metabolic Disease, Institut de Recherches Internationales Servier, 50 Rue Carnot, 92284 Suresnes Cedex, France. [7] The Francis Crick Institute, 1 Midland Rd, London NW1 1AT, UK. [8] These authors contributed equally: Amand F Schmidt, Chris Finan. ✉email: amand.schmidt@ucl.ac.uk

M endelian randomisation (MR) studies estimate the causal relationship of a risk factor of interest to disease outcomes using genetic variants as instruments to index the risk factor. The naturally randomised allocation of genetic variation at conception limits the potential for confounding, which compromises causal inference drawn from the directly observed association between risk factor and disease[1].

Risk factors of interest (some of which are amenable to modification by drugs or behaviour change) can be both exogenous and endogenous, encompassing health-related behaviours (e.g. smoking and alcohol consumption), complex biological traits (e.g. blood pressure and body mass index) or circulating constituents of the blood e.g. complex analytes such as lipoproteins, metabolites such as uric acid, or proteins such as interleukin-6[2–4] (protein quantitative trait loci; pQTL). Interest has also emerged in tissue-level mRNA expression as an exposure of interest[5] (expression quantitative trait loci; eQTL).

Most prior MR analyses utilise an approach whereby multiple SNPs identified from GWAS are used as instruments to increase precision. SNPs are drawn from throughout the genome, often with a single variant selected per locus ensuring instruments are independent (i.e. in linkage equilibrium, LD); preventing erroneously statistical significance. This standard approach has been applied regardless of the position of the exposure of interest in the biological pathway connecting genetic variation to disease risk.

As an example, prior work used MR analysis to assess the causal relevance of the major circulating lipid fractions (LDL-C, HDL-C and triglycerides) for coronary heart disease (CHD), utilising variants from throughout the genome as instruments. However, the approach we explore in this paper uses genetic instruments restricted to the target of interest (acting in cis). The two approaches are both relevant but seek to answer different questions. Whereas the former addresses the causal relevance of the biomarker for CHD, the latter seeks to address whether modification of a specified drug target will reduce CHD and uses the biomarker as a proxy of protein concentration and activity. These two approaches may yield different estimates when the protein drug target, for example CETP, affects multiple pathways (e.g. HDL-C and LDL-C) through so-called post-translational pleiotropy (defined below).

Implicitly cis-MR analyses of drug targets such as CETP, attempt to understand the involvement of the encoded protein in a disease. When proteins of interest are potentially druggable (as with CETP) such MR analyses can be referred to as 'drug target MR'. Recent technological developments enable measurement of hundreds or thousands of proteins on an –omics scale in a single biological sample[4]. This opens up the possibility of scaled drug target MR analysis of thousands of proteins against hundreds of diseases to inform understanding of their causes and improve drug development yield.

In the current manuscript we develop a mathematical framework for drug target MR, showing why these analyses are more robust than MR analyses of more distal traits. We next discuss the choice of exposure variables (mRNA, protein or downstream biomarker), indicating that proteins may be preferred as exposures but when unavailable that drug target MR analyse weighted by mRNA or downstream biomarkers may provide valid test of a protein' effect on disease. Next, we introduce four positive control loci that encode targets of licensed or clinical phase drugs (HMGCR, PCSK9, NPC1L1, and CETP), and empirically evaluate instrument selection strategies, proposing a scalable approach to maximise power while safeguarding against erroneous significance. We further show that tissue-specific drug target MR estimates of eQTL weighted analyses do not always agree with that of similar blood-based pQTL analyses or with evidence from drug trials. Finally, the scalability of our approach is showcased in a phenome-wide scan of an additional five targets on 35 therapeutically relevant outcomes.

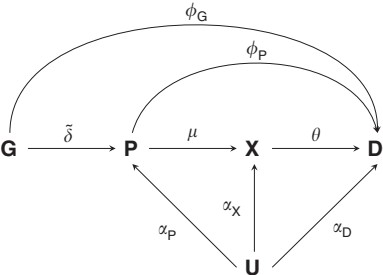

**Fig. 1 Directed acyclic graphs of potential Mendelian randomisation pathways.** Nodes are presented in bold face, with **G** representing a genetic variant, **P** a protein drug target, **X** a biomarker, **D** the outcome, and **U** (potentially unmeasured) common causes of both **P**, **X**, **D**. Labelled paths represent the (causal) effects between nodes.

## Results

**A mathematical framework for *cis*-MR analysis**. MR studies determine the causal effect of a risk factor on a disease using instrumental variable (IV) methods[6], leveraging two estimates: the genetic association with the risk factor (exposure) and the genetic association with the disease (outcome). For the effect estimate in MR to equate to a causal estimate the following critical assumptions should hold: (i) the genetic instrument is (strongly) associated with the exposure, (ii) the genetic instrument is independent of observed and unobserved confounders of the exposure-outcome association (which is secure because genetic variants are fixed and allocated at random), and (iii) conditional on the exposure and confounders, the genetic instrument is independent of the outcome (i.e. there is no instrument—outcome effect other than through the exposure of interest—the 'no horizontal pleiotropy' assumption).

The no horizontal pleiotropy assumption is violated when there are additional pathways by which the instrument may be related to the disease, sidestepping the exposure of interest. In contrast, the association of a genetic instrument with exposures that lie in the causal chain distal to the exposure of interest (vertical pleiotropy[7]) does not violate the assumptions underpinning MR analysis. When proteins serve as the exposure of interest in MR analysis, it becomes possible to give biological context to the concepts of vertical and horizontal pleiotropy. This is because horizontal pleiotropy equates to pathways from gene to disease, which precede translation of the protein of interest, e.g. through alternative splicing or micro-RNA effects. By contrast, vertical pleiotropy refers to the downstream actions of the translated protein, which should be reproduced by a drug with specific action on the protein. Therefore, in the context of MR analysis of proteins, vertical and horizontal pleiotropy correspond to 'pre-' and 'post'-translational effects, respectively. In the methods section we describe how to analytically explore and adjust for horizontal pleiotropy.

Figure 1 illustrates the MR considerations, where a genetic variant (**G**) influences the disease risk (**D**) directly ($\phi_G$) or through its effect ($\tilde{\delta}$) on a protein (**P**), which exerts its action through a downstream biomarker (**X**), which in turn influences disease risk. The relevant genetic effects can be resolved as follows:

(1) The genetic effect on the protein $\tilde{\delta}$.
(2) The genetic effect on a downstream complex biomarker $\tilde{\delta}\mu$.
(3) The genetic effect on disease $\phi_G + \tilde{\delta}(\phi_P + \mu\theta)$, which comprises:

(a) A *direct effect* of the variant on disease $\phi_G$.
(b) An *indirect effect*: $\tilde{\delta}(\phi_P + \mu\theta)$, which is a function of the genetic effect on a protein $\tilde{\delta}$, the direct effect of the protein

on disease $\phi_P$, the effect of the protein on a biomarker $\mu$, and the biomarker effect on disease $\theta$.

Depending on the risk factor of interest, an MR analysis constitutes a simple quotient of the genetic effect on disease by the genetic effect on the risk factor. For example, if we are interested in the causal effect the biomarker $X$ on disease, i.e. $\theta$, we use the following ratio.

$$\frac{\phi_G + \tilde{\delta}(\phi_P + \mu\theta)}{\tilde{\delta}\mu}. \quad (1)$$

For this expression to equate to the causal effect on disease we need to additionally assume that there is no horizontal pleiotropy, in other words $\phi_G = \phi_P = 0$, which reduces the expression to:

$$\frac{\phi_G + \tilde{\delta}(\phi_P + \mu\theta)}{\tilde{\delta}\mu} = \frac{\tilde{\delta}\mu\theta}{\tilde{\delta}\mu}, \quad (2)$$
$$= \theta.$$

In contrast, if we are interested in the causal effect of the protein $P$ on disease $D$, we want to obtain an estimate of $\omega$, where $\omega = \phi_P + \mu\theta$, and assuming $\phi_G = 0$ we calculate the ratio:

$$\frac{\tilde{\delta}(\phi_P + \mu\theta)}{\tilde{\delta}} = \phi_P + \mu\theta, \quad (3)$$
$$= \omega.$$

Critically, where the causal effect of the protein is the parameter of interest, we only need to assume that there is no direct effect of the genetic variant on disease, i.e. $\phi_G = 0$, and the protein can have any mixture of direct ($\phi_P$), and indirect ($\mu\theta$) effects. For this reason, MR analysis of protein-disease relationships is less prone to violation of the 'no horizontal pleiotropy' assumption than MR analysis of downstream exposures. In the methods section we further elaborate on the statistical inference of drug target MR analyses using genetic associations with up- or downstream proxies of a (protein) drug target. We show that in most cases the estimand is distinct from $\omega$. However, under the same no pre-translational pleiotropy assumption, statistical tests will correctly reject the null-hypothesis $\omega = 0$, without inflated false positive rates. In addition, we prove that in the presence of post-translational pleiotropy drug target MR remains valid, even when a downstream proxy (such as lipids) does not causally affect disease.

**Selection of loci encoding proteins.** Unlike MR analysis of non-protein traits, where it has become common to select instruments from throughout the genome, *cis*-MR analysis necessitates the selection of genetic variants from within or near a protein-coding gene. The Ensembl 97 GRCh38 human genome assembly contains an estimated 20,454 protein-coding genes, encoding an estimated 24,700 protein-coding transcripts (merged Ensembl/Havana annotation). Of these transcripts, 21,869 have a degree of experimental support for the presence of these transcripts. In addition, UniProt (version 2019_06, combining SwissProt and TrEMBL) reports 20,416 high quality manually annotated proteins.

**Selection of loci encoding druggable genes.** Not all encoded proteins are amenable to pharmacological action by (small molecule) drugs, or peptide and monoclonal antibody therapeutics, which currently account for the majority of medicines. *cis*-MR for drug target validation requires the selection of genes encoding druggable proteins. Progressive efforts to delineate the druggable genome[8,9] (available through the DGI database (DGIdb[10]), have culminated in the latest iteration containing

4,479 genes[11] encompassing targets of existing therapeutics, potentially druggable close orthologues and targets accessible by monoclonal antibodies. Of these, ChEMBL v.24 identifies 896 genes as encoding the target components for existing therapeutics. These include single protein targets, protein complex targets and targets comprising whole protein families. A further 535 genes encode target components of compounds currently in clinical testing. The druggable genome is not static and will be redefined periodically, reflecting changes in drug targeting mechanisms. However, currently, to define the druggable genome is to progressively reduce the high-dimensional search space for genetic instruments from the whole genome to around 20,000 protein-coding genes to fewer than 5000 genes encoding druggable targets. As such, a specific subset of *cis*-MR can inform drug development, which we term 'drug target MR'.

**Instrument selection.** Drug target MR focuses on a gene known to encode a protein drug target, and variants within and around a gene are used to characterise the effect(s) of the drug target on a single or multiple outcome(s). Given the inferential target, it would seem logical to select variants based on the variant to protein level association ($\tilde{\delta}$) in a relevant tissue. Ideally one would only select causal variants known to affect the drug target, maximising precision (power). However, typically the nature and number of causal variant(s) is unknown, imposing the need for instrument selection. Similarly, while the first GWAS on the proteome are becoming available, currently most drug target MR analyses utilise biomarkers downstream of the drug target protein. As such, to evaluate the current modus operandi, we explore instrument selection using biomarker proxies, and compare this to using the actual pQTL effects, and alternatively eQTL effects (as upstream proxies of a drug target).

In such cases, variants are often selected based on (1) a biomarker association (e.g. LDL-C in the case of PCSK9 discussed earlier), (2) predicted functionality; and (3) low LD, typically including a single[12,13] or perhaps a handful of SNPs[14,15], out of a multitude of potential candidate SNPs. It is often unclear how well such a small subset of SNPs characterises the drug target effect, and how influential such strategies are on the final result.

To explore this, we mimicked instrument selection by repeatedly (500 times) sampling four SNPs at random per locus from four known drug target encoding loci *HMGCR* (statins), *NPC1L1* (ezetimibe), *PCSK9* (PCSK9 inhibitors), and *CETP* (CETP inhibitors). These loci contain variants that influence LDL-cholesterol (*HMGCR*, *NPC1L1*, *PCSK9*, *CETP*) with variants at the CETP locus additionally influencing HDL-cholesterol and triglycerides as identified by the Global Lipids Genetics Consortium (GLGC[16]). We used a generalised least squares (GLS) method[17,18] to account for pairwise LD between variants at each locus. Variants were extracted from within the gene ±2.5 kB, with a minor allele frequency (MAF) above 0.01, and LD <0.80 (Supplementary Tables 1–5, Supplementary Fig. 1).

The first and third quartiles (Q) of the CHD odds ratios (OR) per standard deviation (SD) in LDL-C for *HMGCR*, *NPC1L1*, *PCSK9* (or HDL-C in the case of *CETP*) indicated modest variability in the point estimate: (Q1 1.61, Q3 1.78) for *HMGCR*, (Q1 1.42, Q3 1.77) for *PCSK9*, (Q1 1.19, Q3 1.68) for *NPC1L1*, and (Q1 0.87, Q3 0.91) for *CETP*. Between 95 and 99% of the estimates across all four genes were in the expected direction as inferred from the findings of drugs used in clinical trials to target the corresponding proteins[19–25].

We further categorised effect estimates based on the EnsEMBL Variant Effect Prediction (VEP)[26] of each variant (Fig. 2), finding little to no difference between estimates derived using non-coding

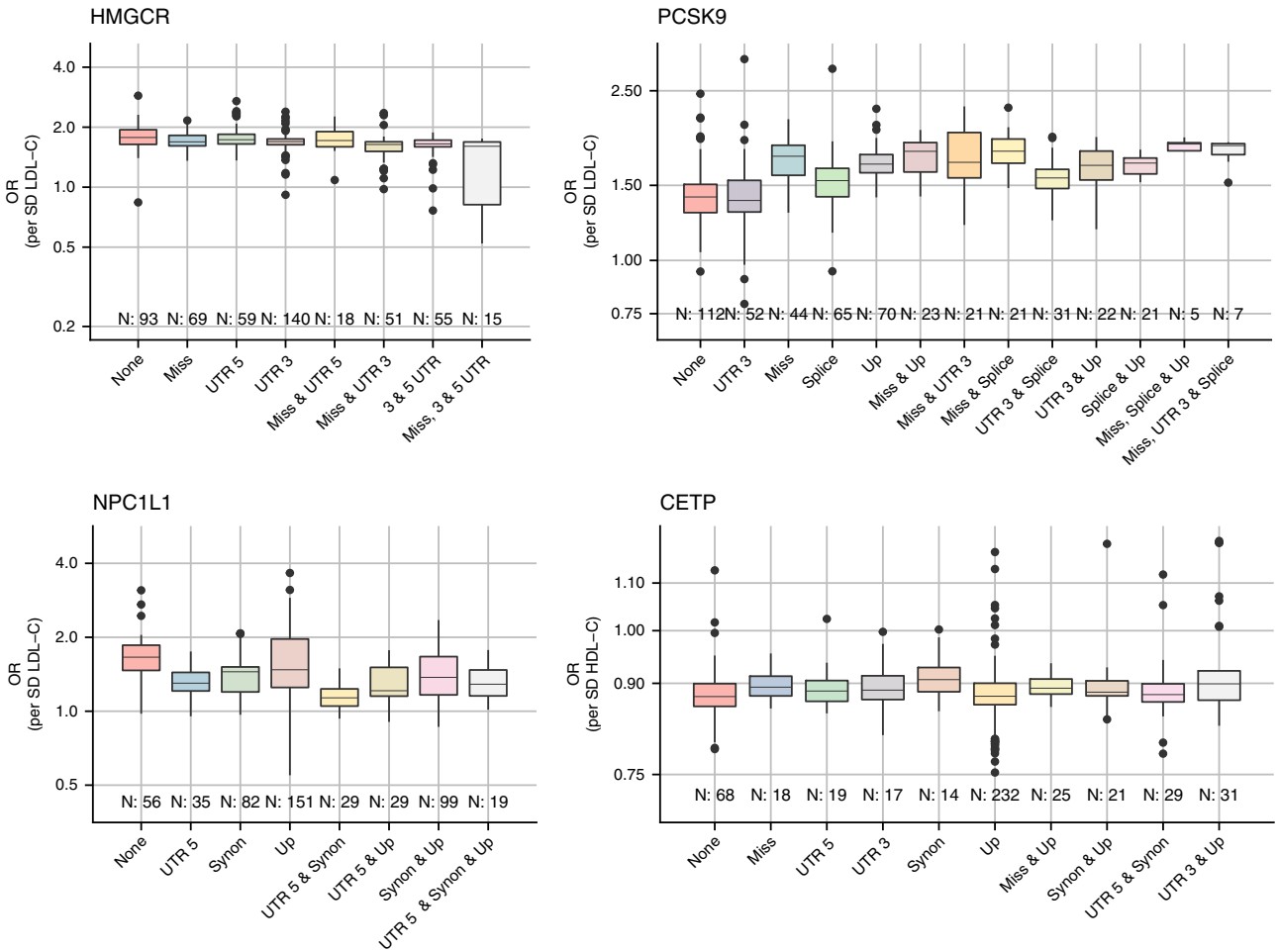

**Fig. 2 Instrument selection related variation in the point estimates of drug target Mendelian randomisation studies on the lipid's association with CHD.** Each estimate is based on randomly (500 iterations) selecting 4 SNPs out of 17 *HMGCR*, 30 *PCSK9*, 21 *NPC1L1*, 36 *CETP* candidate variants. Lipids data were used from the GLGC and linked to coronary heart disease data from CardiogramPlusC4D. estimates were grouped by the inclusion of instruments with worsted predicted functional or regulatory consequence; categories occurring less than five times were removed. Any pairwise LD was accounted for using the 1000 genomes 'EUR' reference panel and a generalised least squares method[17]. The boxplots depict quartiles 1, 2 (median), and 3 as a box, with the whiskers presented as vertical bars and values ±1.5 times the interquartile range as dots.

variants only and those estimates based on functional variants. The overall stability and agreement between estimates derived with or without functional annotations suggests a strong influence of multivariate LD between the selected and unselected variants in small *cis*-regions (Supplementary Fig. 1). We did however observe a large degree of variability in the *p* values which is explored in the subsequent section (Supplementary Fig. 2).

**Taking advantage of linkage disequilibrium within the region.** Given the observed influence of LD it seems desirable to leverage this in drug target MR. For example, after defining a *cis*-genetic region (discussed further below) one can LD-clump highly correlated variants that might destabilise a statistical model (multicollinearity) and account for remaining pairwise LD using an LD-reference panel to maximize power and decrease variability. Besides increasing power and robustness, this strategy introduces some complexities, e.g. the choice of LD threshold.

The effect of LD thresholds can be readily explored by performing a 'grid search', clumping variants at different R-squared thresholds. From modelling theory (and empirically: Fig. 2), one would expect that when using such a grid search the point estimate stabilises early (at low thresholds), while the standard errors decrease further until, at a certain point,

multicollinearity results in a clear deviation from the overall tendency. Such a grid search was implemented in Fig. 3 showing signs of multicollinearity for the *HMGCR* and *PCSK9* estimates, but less so for *NPC1L1*. While trends observed for *HMGCR* and *PCSK9* are examples of what one would expect on theoretical grounds, this does not occur at the same threshold, and seemingly not at all for the *CETP* locus. Given the ongoing debate on whether the beneficial effect of CETP-inhibition depends on HDL-C raising or LDL-C lowering activity, we repeated these analyses using LDL-C weights (Supplementary Fig. 3) with similar results to those observed using HDL-C weights.

**Linkage disequilibrium modelling compared with selecting functional variants.** Based on these considerations, we explored the performance of a very limited instrument selection strategy, geared towards characterising a *cis*-genetic region encoding a drug target as fully as possible by (1) considering all variants with limited LD clumping to prevent multicollinearity; (2) selecting the most significant variant in an 'LD-block'; (3) modelling LD using external data such as the 1000 genomes reference panel. This strategy (with $R^2 = 0.60$) was applied to our four empirical examples and compared with MR estimates at the same locus using only variants with strong evidence of function based on

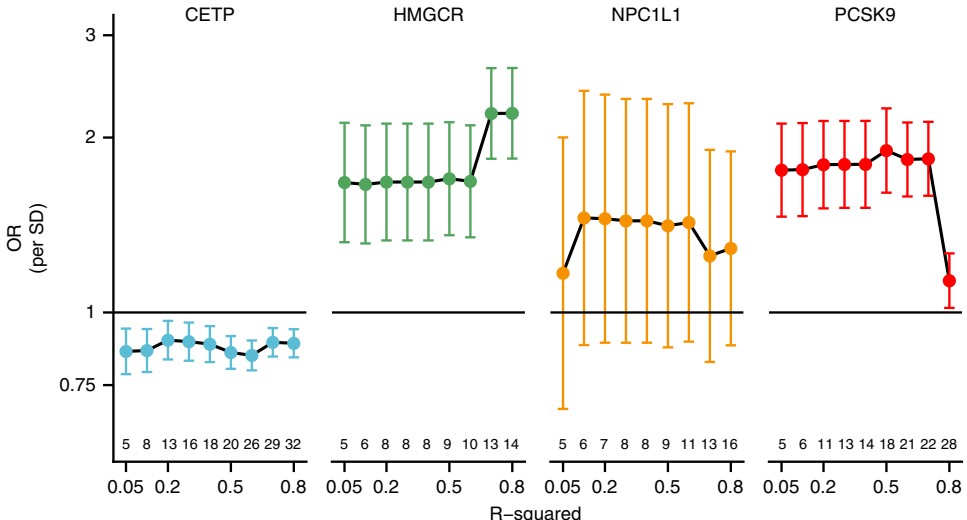

**Fig. 3 Mendelian randomisation estimates of the lipids weighted associations with CHD under increasingly liberal LD-clumping thresholds.** Lipids data were used from the GLGC, and linked to coronary heart disease data from CardiogramPlusC4D. Pairwise LD remaining after LD-clumping was accounted for using the 1000 genomes 'EUR' reference panel[52] and a generalised least squares method[17]. Estimates for PCSK9, HMGCR, and NPC1L1 are given per SD in LDL-C, CETP estimates per HDL-C reflecting the likely effectiveness pathway to CHD. The number of included variants is depicted above the x-axis. Estimates are given as OR with 95%CI (vertical error bars).

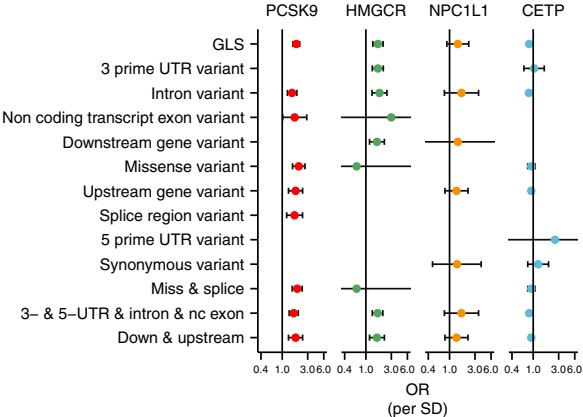

**Fig. 4 Mendelian randomisation estimates of the lipids weighted associations with CHD stratified by functionally of the included variants.** Lipids data were used from the GLGC, and linked to coronary heart disease data from CardiogramPlusC4D. Pairwise LD remaining (after clumping on R-squared of 0.60) was accounted for using the 1000 genomes 'EUR' reference panel[52] and a (GLS) generalised least squares method[17]. Estimates for PCSK9, HMGCR, and NPC1L1 are given per SD in LDL-C, CETP estimates per HDL-C reflecting the likely effectiveness pathway to CHD. Estimates are given as OR with 95%CI (vertical error bars). Numerical details, including the number of variants used, are provided in Supplementary Tables 6-9.

VEP (Fig. 4 and Supplementary Tables 6–9). We found general agreement between the effect estimates from both analytical approaches, with the GLS estimates having higher precision (1/ SE) than those based on functional variants alone. For example, the precision of the two *PCSK9* splice variants used in MR was 5.72 compared with 13.33 for the GLS estimates (incorporating variants selected based on LD structure irrespective of function).

These results confirm that precision/power is increased by including more correlated variants. To prevent erroneously low *p* values in such analyses, we accounted (conditioned) for pairwise LD using the European 1000 genomes panel. We further investigated the influence of different 1000G ancestry reference panels on the effect estimates and found these to be stable for the four examples evaluated (Supplementary Fig. 4); although significance of the *NPC1L1* was dependent on the panel used. We did, however, observe that the GLS method often failed because of (small) changes in LD resulting in multicollinearity. After inspection, this seemed to be related to LD-estimates of low MAF variants varying across ethnicities (Supplementary Fig. 5). Improved behaviour may be expected with either increased sample size (1000G sample size *n*~100), or with the removal of low MAF variants (at the risk of losing information).

**Drug target MR using pQTLs.** The analyses to this point utilised lipid exposures to index the effect of four drug targets. Large scale GWAS studies of the proteome have recently become possible (e.g. the INTERVAL study[4]), opening up the possibility of using the genetic effect on protein concentration as a more direct proxy of the drug target effects. Of the four proteins considered thus far, we had access to pQTL estimates from GWAS of circulating CETP and PCSK9 concentration measured by enzyme-linked immunosorbent assays (ELISA)[27,28] in about 4000 and 3000 subjects, respectively. Initially focussing on variants in the same ±2.5 kB region as before, we found circulating CETP increased CHD risk, consistent with the findings of a recent large-scale clinical trial where CETP inhibition reduces CHD risk (Fig. 5). Similarly convincing results were observed for the analyses of log(PCSK9) concentration and CHD.

In the biomarker weighted analysis, the size of the genetic flanking region was constrained to prevent erroneously modelling effects from neighbouring genes not encoding the drug target of interest (horizontal pleiotropy). However, pQTL associations provide a direct estimate of the genetic association with the drug-target and hence may reduce the need to focus on small flaking regions. We therefore compared findings from the ±2.5 kB region to pQTL MR results using a broader ±1 MB flanking region. To further guard against potential horizontal pleiotropy (for example through LD) we additionally implemented the Egger adjustment. CETP was (again) robustly causally associated with CHD, with larger R-squared values decreasing variability without any indication of model instability (Fig. 5). Due to the limited number of additional variants, the ±1 MB PCSK9 pQTL analysis

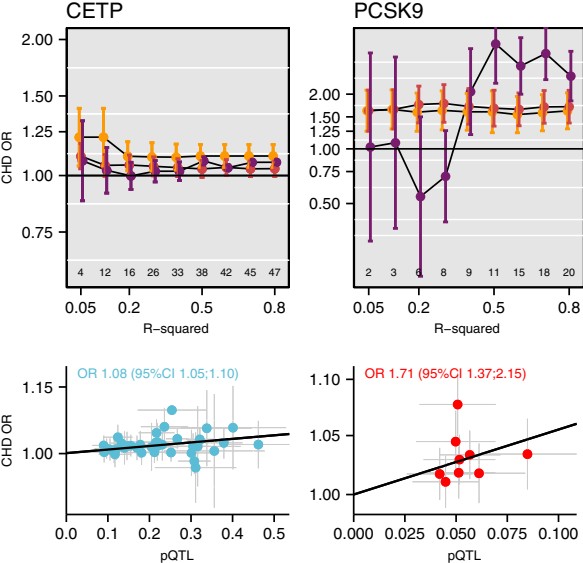

**Fig. 5 Mendelian randomisation estimates of protein level effects on CHD, with a grid of LD threshold.** Pairwise LD was accounted for using the 1000 genomes 'EUR' reference panel[52] and a (GLS) generalised least squares method[17] with or without Egger correction for possible horizontal pleiotropy. The number of included variants in the 1 mega base flanking region is depicted above the x-axis of the top panels. Estimates are given as OR with 95%CI (vertical error bars). The top panel depicts the variant to CHD or protein level effect for clumping threshold 0.5 for CETP (based on an Egger correct GLS model), and at 0.4 for PCSK9 using an IVW GLS model; 38 and 9 variants, respectively. Notice that the PCSK9 estimates were only available on the natural logarithmic scale.

did not differ markedly from the ±2.5 kB, and the MR-Egger estimate was fairly unstable.

We further performed eQTL weighted analyses of the same positive control drug targets, observing persistent directional discordance between the tissue specific CHD association of a single drug target. For example, PCSK9 mRNA expression in the adrenal gland was associated with an increase in CHD risk: OR 1.09 (95%CI 1.02; 1.16), while PCSK9 expression in the uterus was associated with decreased CHD risk: OR 0.92 (95% 0.88; 0.97). This discordance was not explained by horizontal pleiotropy, region size, or influence of enhancer variants (Supplementary Discussion, and Supplementary Tables 9,10).

**Further examples: phenome-wide drug target analysis**. To further showcase the generalisability of the proposed approach, we conducted drug target analyses focussing on circulating proteins that are the direct efficacy targets of clinical phase developmental or licensed drugs. Genetic instruments for cis-MR analyses were identified from the INTERVAL study which conducted a GWAS of around 3000 circulating proteins[4], measured using the Somalogic aptamer-based proteomics platform. From the reported data we identified five proteins encoded in the druggable genome[11] (*F10, IL-12, PLG, IL-1R1, MMP-9*) for which sufficient sentinel variants could be identified. We then conducted a phenome-wide drug target MR analysis against 35 clinically relevant disease and biomarker phenotypes (see 'Methods').

Circulating factor X (encoded by *F10*) was associated with a higher risk of any stroke (OR 1.13 95%CI, 1.05,1.21), which is keeping with the use of direct-acting anticoagulant drugs that inhibit factor X (e.g. apixaban) to prevent stroke in patients with atrial fibrillation (AF)[29]. Furthermore, we found a possible effect

of factor X on asthma (OR 0.78 95%CI 0.62, 0.99) (Fig. 6). The monoclonal antibody ustekinumab directed against a common subunit of interleukin 12 and interleukin 23 interferes with the binding of these cytokines with the IL-12 receptor to inhibit inflammatory signalling[30]. Ustekinumab has European marketing authorisation for the treatment of psoriasis and Crohn's disease (CD) after demonstrating efficacy in clinical trials[31], and is under evaluation for ulcerative colitis (UC). Consistent with this, genetically instrumented higher interleukin-12 subunit beta (encoded by *IL12B*) was associated with a higher risk of CD (OR 2.02; 95% CI 1.48, 2.76), UC (OR 1.56; 95% CI, 1.31, 1.87), and inflammatory bowel disease (IBD) (OR 1.56 95%CI 1.31, 1.87) (Fig. 6). Genetically-instrumented higher circulating concentration of plasminogen (encoded by *PLG*) was associated with a lower ischaemic stroke risk (OR 0.85; 95% CI 0.72, 1.00) (Fig. 6), consistent with the known efficacy of recombinant tissue plasminogen activator (tPA) for acute ischaemic stroke[32]. The PLG association with an increased risk of any stroke (OR 1.06; 95%CI 1.01, 1.11) is presumably due to an increase in haemorrhagic events. Increased levels of PLG were furthermore associated with increased risk of AF, IBD, CD, Alzheimer's disease (Fig. 6) as well as lipids, and increased SBP (Fig. 6), but these effects may not be observed therapeutically because tPA is given as a single dose in acute MI and ischaemic stroke. Higher circulating concentration of interleukin-1 receptor 1 (encoded by *IL1R1*) was associated with a lower risk of both CD, IBD and UC. This would be in keeping with the circulating form of the receptor functioning as decoy to reduce signalling through the membrane-bound form of the receptor by the pro-inflammatory cytokine interleukin 1β. We also found evidence through cis-MR of a causative role for MMP9 in CD and IBD. Recent phase 2 trials failed to demonstrate efficacy of andecaliximab, a monoclonal antibody targeting MMP9 in either UC or CD[33,34] (Fig. 6). However, given the evidence from the MR analysis, further consideration should be given to the type, dose, frequency and duration of ant-MMP9 therapy in Crohn's disease before this target is discounted for these diseases.

## Discussion

We have used biological and mathematical arguments to formalise the distinction between locus-specific Mendelian randomisation (MR) analysis for drug target validation, where the appropriate instruments are variants in or within the vicinity of the encoding genes, and other types of MR analysis, e.g. for risk factor validation, where instruments are used from throughout the genome. Using algebraic derivations, we show that because drug target MR considers the effects of perturbing a protein drug target on disease, this type of MR may be applied in settings where traditional MR, focussed on distal traits, could be biased through horizontal pleiotropy. We also illustrate the challenges when undertaking MR for drug target validation. These include defining the loci of interest, accounting for linkage disequilibrium, and selecting the exposure through which to weight the genetic outcome association.

We discuss resources available for the identification of 'druggable' protein-coding genes and show that, because MR for drug target validation is framed as a cis-focused analysis, instrument selection is distinct from that for MR for validating the causal relevance of a non-protein or environmental exposure. We investigated strategies for characterising the drug target-encoding region through linkage disequilibrium (LD). Grid-search algorithms were introduced aiding researcher in optimising LD-thresholds, as well as genetic regions, with intuitive sensitivity analyses to estimate robustness to the choices of LD reference panel, the presence of functional variants as well as regulatory

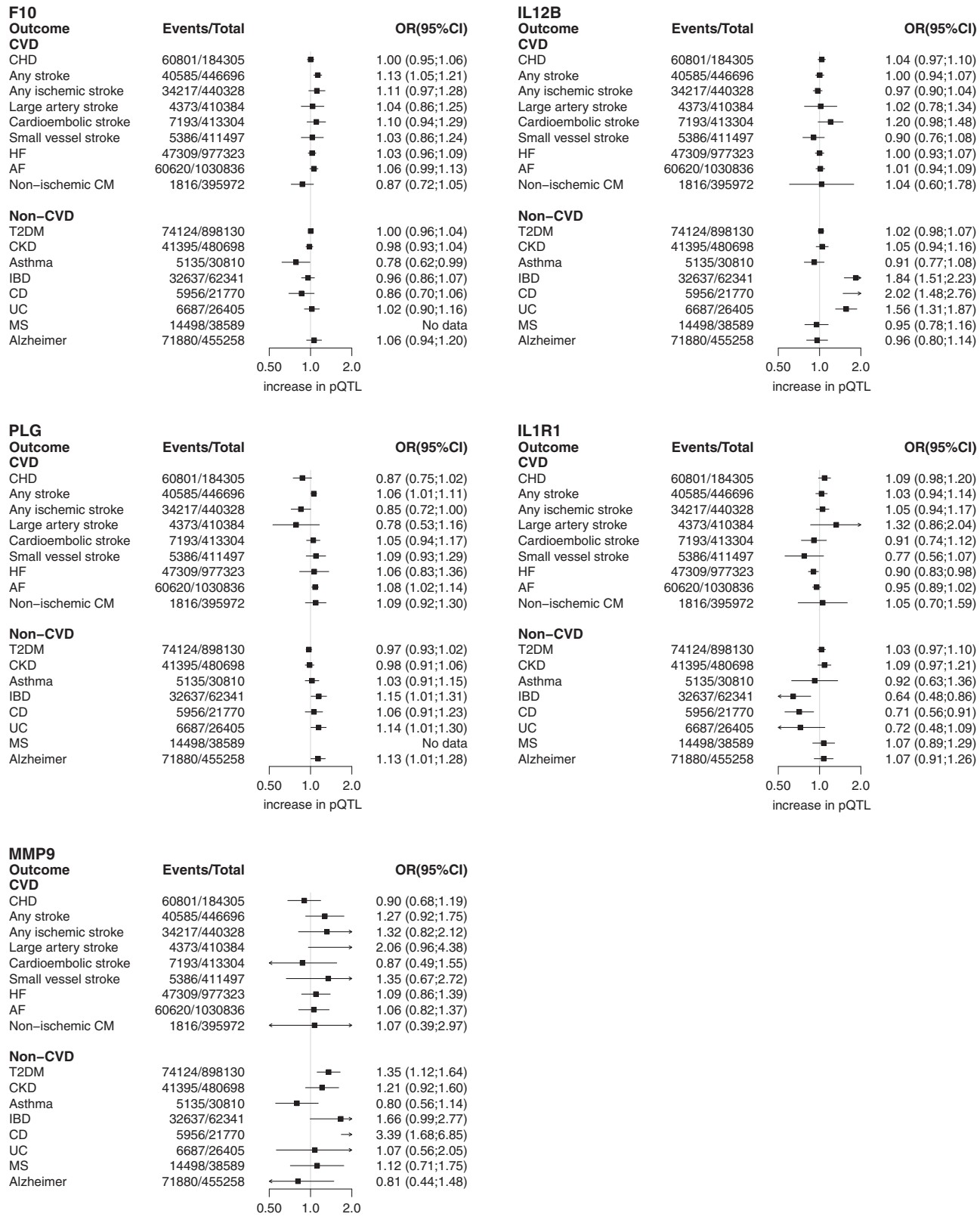

**Fig. 6 A pQTL based drug target MR phenome wide scan.** Results are presented as odds ratios with 95% confidence intervals (horizontal lines), positioned in the protein increasing direction. The total number of events and sample size are provided in the forest plot.

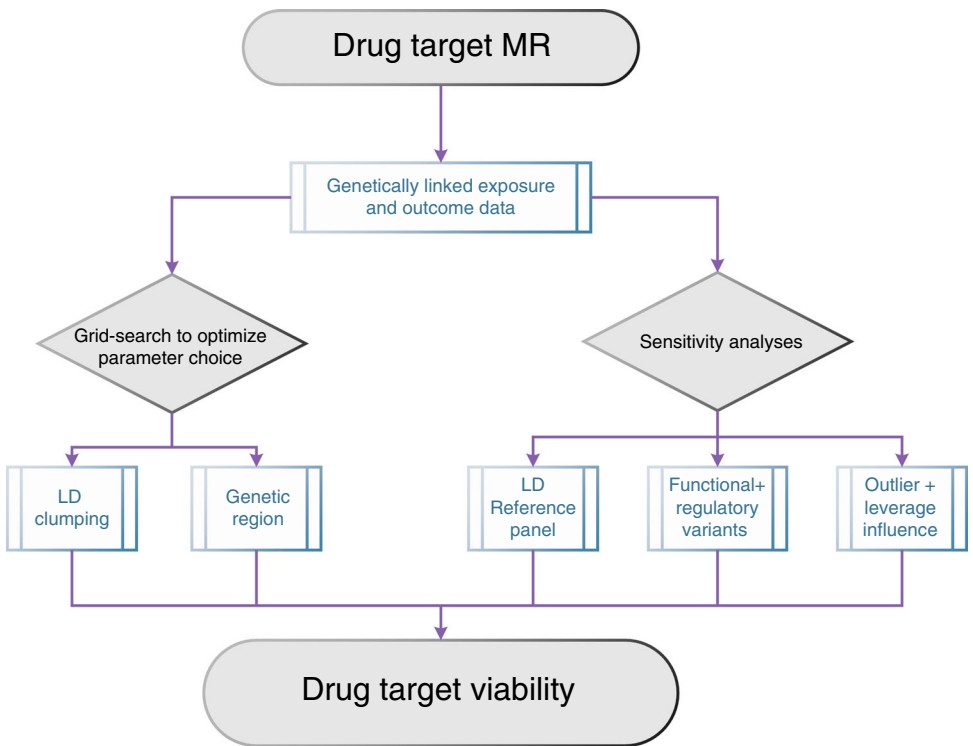

**Fig. 7 A proposed drug target MR analysis framework.** The influence of LD and genetic region can be explored (and optimized) through simple grid-searching. Robustness of model choices in LD reference panel, the selection of functional or regulatory variants, and outlying or influential (high leverage) variants can be explored thorugh sensitivity analyses showcased here and in the supplementary analyses.

enhancers, and outliers using heterogeneity statistics. We further introduce an exploratory analysis to determine the influence of horizontal pleiotropy, pruning variants that associate with other genes around a locus encoding a drug target (see Methods section and Supplementary Information). Finally, we illustrate the generalisability of our approach by presenting findings from a pQTL drug target phenome-wide scan across 35 clinically relevant outcomes.

Due to our focus on four positive control examples, we were able to perform exhaustive analyses on the robustness of drug target MR findings based on regulatory versus coding variants showing that robust causal inferences could be drawn from regulatory variants despite a widely held view that functional variants should be naturally preferred in (drug target) MR. Instead we showed that applying a straightforward clumping algorithm, agnostic of the type of variants, resulted in the same OR, compared with selecting functional variants, with greater precision (increased power) by including larger numbers of (partially dependent) variants. Similarly, we showed the limited influence of LD-reference panels used in LD-modelling with non-European ancestry panels resulting in comparable estimates. While promising, these findings should be replicated and above all extended to a larger set of drug targets, for example to analyse targets outside of the lipids-cardiovascular domain presented here.

While there are an impressive number of estimator functions (see methods), there has been limited advice on instrument selection for drug target MR, with sparse attention given to empirically exploring the influence on MR estimates. This manuscript addresses both issues and introduces a generic framework (Fig. 7) for obtaining robust inference of a drug target' effect on disease, irrespective of the type of MR estimator method preferred. In this framework we suggest that for each exposure outcome pair grid-searches are employed to select the optimal LD threshold and genetic region, while at the same time exploring the

robustness of MR estimates to LD-reference panel, the influence of functional and regulatory variants, as well as assessing the influence of outlying instruments.

MR of protein exposure have been conducted before, sometimes selecting both *cis* and *trans* and sometimes selecting only *trans* variants. However, the use of *cis* instruments for MR analysis of proteins is less prone to violation of the horizontal pleiotropy assumption than the use of *trans* instruments (see Supplementary Methods) and is amply illustrated by the example of C-reactive protein (CRP). Circulating CRP concentration is associated with CHD risk in nonrandomized observational studies. In 2011, in a paradigm example, a *cis*-MR analysis of CRP in CHD[35] showed the nonrandomized association is non-causal. Consistent findings were obtained by others[36]. Subsequent GWAS of CRP have been conducted[37] and have identified variants in genes outside CRP (acting in *trans*) that associate with CRP concentration, including in genes encoding the receptor for the inflammatory cytokine IL-6 and *APOC1*, involved in lipid metabolism. Variants in *IL6R* that are associated with lower CRP concentration are associated with lower risk of CHD[2,3] and variants in *APOC1* that are associated with higher concentration of CRP are associated with increased risk of CHD[38]. However, it would be erroneous to suppose that a *trans*-MR analysis of CRP instrumented using *APOC1* variants provides evidence of a causal role of CRP in CHD[38] since the same variants are also associated with LDL-C. Clearly, leveraging variants in *APOC1* (or *IL6R*) acting in *trans* to probe the causal relevance of CRP for CHD would introduce bias through horizontal pleiotropy. The same argument applies to the use of variants in *IL6R* for the same purpose. Signalling through the interleukin-6 receptor encoded by *IL6R* influences many inflammatory molecules beyond CRP that are the likely mediators of its effect on CHD.

Over 90% of drug targets are proteins, therefore weighting by protein expression in a disease-relevant tissue would provide the

most informative *cis*-MR analysis for drug target validation. Many of the circulating proteins measured by the currently available proteomics platforms (e.g. from Somalogic [2036 druggable proteins] or O-link [973 druggable proteins]) are the actual biological efficacy targets of many licensed or developmental therapeutics. Thus, the available data on the circulating proteome already provide a step change in the ability to apply genetic target validation. We have illustrated the potential of this approach by conducting a *cis*-MR and PheWAS for five such targets.

Where drugs bind membrane bound or intracellular proteins, which subsequently affect non-protein constituents of the circulating blood, drug target MR studies might fruitfully employ genetic associations with downstream proxies of protein level and concentration. We illustrated this with reference to four drug targets for lipid-lowering (HMGCR, NPC1L1, PCSK9 and CETP). For two of these targets (PCSK9 and CETP) it was also possible to compare the findings of *cis*-MR analyses weighted by the level of the circulating protein with *cis*-MR analyses weighted by the relevant lipid fraction.

However, for many drugs or drug targets, for example, those used in, or being developed for, the treatment of neurological, myocardial, or musculoskeletal conditions, a circulating biomarker may be unavailable or may not represent a strong proxy for the drug target. Tissue-specific pQTL data have yet to be generated at any scale and, until such data become available, we investigated tissue-specific eQTLs as a potentially relevant alternative exposure that might closely proxy pQTL effects. We found that eQTL-based MR estimates may differ both in magnitude as well as direction across tissues, as demonstrated by exhaustive analyses of the *HMGCR*, *NPC1L1*, *PCSK9* and *CETP* loci. This tissue-dependent heterogeneity was independently reported by the GTEx consortium for *PCSK9*[39]. We extend those observations to demonstrate their potential to undermine reliable causal inference when using mRNA expression as a weighting variable in MR analysis. Possibly, the observed heterogeneity may relate to the inclusion of non-European ancestries in the GTEx database, or due to the post-mortem collection of samples[40]. For example, GTEx previously reported that gene expression changed post-mortem in a tissue-specific manner, which they attempted to ameliorate through multiple regression[40].

This tissue-specific heterogeneity likely reflects actual biology which might also extend to tissue-specific pQTL data. A key uncertainty is identifying the 'relevant' tissue for a drug target validation MR analysis. This would be informed by greater knowledge of the protein expression profiles of all human drug targets, an area that has so far received limited attention. When these data become available, it will become important to evaluate tissue dependency and the underlying mechanisms in more detail. Where it is relevant, and possible, to use circulating proteins or biomarkers such as lipids as exposure variables in two-stage drug target MR analysis this may help mitigate the complexity of weighting based on tissue-specific eQTL or pQTL data.

In this current manuscript we have exclusively focussed on MR as a tool for drug target validation, however many complimentary methods exist, often utilising non-genetic cell, tissue and animal experiments. A key challenge to further improve (early) drug development will be to incorporate these different sources of evidence to accurately predict in-human efficacy.

In conclusion, we expect that combining the discussed concepts with the ever-increasing magnitude of genetic and other 'omics' data will move drug target MR from manually curated, often proof of concept-like analyses, to more automated and scalable projects able to systematically guide and enrich the entire drug development process.

## Methods

**Mendelian randomisation for protein exposures.** There are reasons for considering MR analysis of a protein drug target to be a distinct category of MR analysis. First, an analysis of this type induces a natural dichotomy in the genetic instruments that might be used: those that are located in and around the encoding gene ('*cis*-MR') vs those located elsewhere in the genome (*trans*-MR). Second, aside from mRNA expression, differences in protein expression or function are the most proximal consequence of natural genetic variation. This has two consequences: frequently, variants located in and around the encoding gene can be identified with a very substantial effect on protein expression in comparison to other traits; moreover such instruments may also be less prone to violating the 'no horizontal pleiotropy' assumption than variants located elsewhere in the genome (discussed below and in ref. [41]). Third, in the case of MR analysis of proteins, Crick's 'Central Dogma'[42] imposes an order on the direction of information flow from gene to mRNA to encoded protein, which does not extend beyond this to other biological traits that lie more distally in the causal chain that connects genetic variation to disease risk. Finally, *cis*-MR of a protein risk factor greatly reduces the risk of reverse causation, because Crick's dogma indicates that the pathway gene → encoded protein → disease would always be favoured over the pathway gene → disease → encoded protein, especially given that the gene → encoded protein association is typically derived from population-based (disease-free) samples. Thus, from an MR perspective, proteins are in a privileged position compared with other categories of risk factor and the use of *cis*-MR represents an optimal approach to instrument their causal effect for disease (See Supplementary Methods and Supplementary Figs. 7, 8).

The key to realising this potential is the development of a robust conceptual and mathematical framework for *cis*-MR analysis of proteins. Since *cis*-MR analysis restricts selection of genetic instruments to those located in, or near the encoding gene, new questions emerge as how to optimise the selection of such variants. These include how best to select and define the loci of interest, the physical distance around each gene from which instruments might be drawn; how to select genetic variants as instruments with options including 'no selection', 'selection by strength of association', or 'according to functional annotation'. Regulatory, non-coding variants act through the level of the encoded protein which is what high-throughput assays detect. Coding-variants might influence protein activity but may also alter the detected rather than actual protein level by protein epitope changes, resulting in a technical artefact. Further questions include whether to weight such instruments in an MR analysis by the level of protein *expression* or *activity*, where the relevant assays are available; or, where they are not, by the level of mRNA expression (and, if so, in which tissue), or by some downstream consequence of protein action, e.g. differences in the level of a metabolite known to be influenced by the protein such as LDL-C for a lipid-lowering drug target.

**Alternative exposures in *cis*-MR analysis of proteins.** Drug target MR is concerned with obtaining inference on the effect direction of perturbation of a (protein) drug target, where the effect estimate can be defined as $\omega = \phi_P + \mu\theta$ (Fig. 1 and Supplementary Fig. 7). It is important to note that a protein can remain the inferential target in an MR analysis even if it is not measured directly. For example, in cardiovascular disease large sample size GWAS are available on lipids which are often intermediate biomarkers, positioned downstream between the drug target $P$ and disease, $D$ (Fig. 1). In our recent drug target MR analysis of PCSK9[14,43] we used instruments selected from the encoding locus and divided the variant to coronary heart disease (CHD) estimates, not by the effect on PCSK9 level (which was unavailable), but by LDL-C, a variable known to be altered by perturbation of the PCSK9 protein. Thus, using the same notation as above, and assuming $\phi_G = 0$:

$$\omega_{bw} = \frac{\bar{\delta}(\phi_P + \mu\theta)}{\bar{\delta}\mu} = \frac{\phi_P + \mu\theta}{\mu},$$
$$= \frac{1}{\mu} \times \omega,$$

with $bw$ indicating 'biomarker weighted'. Clearly because the denominator contains $\bar{\delta}\mu$, instead of $\bar{\delta}$, $\omega_{bw}$ does not equal $\omega$. However, $\omega_{bw}$ may still provide a valid null-hypothesis test of $\omega = 0$, because $\omega_{bw} \neq 0$ implies $\omega \neq 0$. Notice that if the protein has a direct effect on the disease, that is not mediated by the downstream biomarker, $\omega_{bw} \neq 0$ does not provide evidence for the biomarker itself to causally effect disease; i.e, $\omega_{bw} \neq 0$ does not imply $\theta \neq 0$. The only additional requirement for using a 'biomarker-weighted' MR for drug target validation is that the protein is strongly correlated with the downstream biomarker, for example when $\mu \neq 0$; which is a (slightly) different version of IV assumption (i).

In the absence of available measures of the protein of interest, a similar argument can be made for using mRNA expression (this time as an upstream variable) that proxies the effect of genetic variation on the level of the encoded protein again within the framework of a *cis*-MR analysis (see Supplementary Fig. 7):

$$\omega_{ew} = \frac{\bar{\delta}_{GE}\bar{\delta}_{EP}(\phi_P + \mu\theta)}{\bar{\delta}_{GE}} = \bar{\delta}_{EP}(\phi_P + \mu\theta),$$
$$= \bar{\delta}_{EP} \times \omega.$$

Here the weighting is done by the association with of mRNA expression and the $\bar{\delta}$ effect has been decomposed into the variant effect on expression $\bar{\delta}_{GE}$ and the

expression effect on protein level $\bar{\delta}_{EP}$. Similar as for $\omega_{bw}$, the expression weighted ('ew') drug target effect provides a valid test of $\omega = 0$ conditional on the absence of any pre-translational pleiotropy; that is, a necessary assumption $\phi_G = \phi_E = 0$ (with index $E$ for expression). It should be noted that, similar as for protein expression, mRNA expression level (eQTL) is tissue-specific and utilising eQTLs for drug target MRs necessitates a decision on the tissue(s) relevant for (de novo) drug development.

**Exploring horizontal pleiotropy.** To address the possibility of horizontal pleiotropy in genome-wide MR analyses of risk factors, it has been common to select independent instruments from multiple locations across the genome. In doing so, the average horizontal pleiotropy may reduce to zero (so-called *balanced* pleiotropy). As noted above, drug target MR is often performed using *cis* variants, selected form within and around a drug target encoding locus. While this diminishes the likelihood of horizontal pleiotropy (Supplementary Methods), should horizontal pleiotropy nevertheless occur, due to the proximity of variants, it is more likely to be directional in nature. In such cases estimators such as MR-Egger[44] might recover the causal effect contingent on the **I**nstrument **S**trength is **I**ndependent of the **D**irect **E**ffect (INSIDE) assumption[44]; i.e. that the strength of the genetic association with the risk factor does not correlated with the magnitude of horizontal pleiotropy. Additionally, an individual variants' contribution to the degree of horizontal pleiotropy (and excluded from analysis) can be explored using straightforward metrics for outlier detection such as the Q-statistic (the squared distance from the average estimate), or through leverage statistics (the estimate change after removing a variant).

Due to the *cis* focus of most drug target MRs it is essential to ensure that pre-translational horizontal pleiotropy is absent. For example, a variant associated with PCSK9 expression may also associate (e.g. through LD) with the expression of other genes. Here we introduce a novel empirical method to exclude LD-based horizontal pleiotropy by sequentially pruning eQTL data for an association with the expression of a 'non-target' gene within a certain flanking region (here ±1 MB) of the encoding locus (using $p$ value threshold of $\{10^{-8}, ..., 1 \times 10^{-3}\}$; Supplementary Fig. 9). This screening step is showcased using the eQTL weighted MR analyses discussed in detail in the Supplementary Data section. We note that the type of screening can however be applied irrespective of the intended exposure, for example it could also be used in drug target MR analyses using pQTL exposures.

In general, across the four positive control loci we did not see much influence of LD-based horizontal pleiotropy; and within a single tissue we did not observe much directional discordance. For each drug-target we did observe a few tissue-specific associations that only obtain significance after pruning potential pleiotropic variants to a very low $p$ value threshold. For example, *CETP* expression in the colon is only associated with CHD after removing variants that had a $p$ value $< 1 \times 10^{-3}$ with neighbouring genes.

**Null-variants and weak instrument bias.** In the current manuscript we propose to implement drug target MR through LD clumping of variants and modelling residual correlation using external reference data[45]. A perceived drawback of this approach is the possible inclusion of 'null variants', that do not affect the intermediate risk factor. In a conservative attempt at excluding null-variants researchers often focus on genome-wide significance (e.g. a $p$ value $< 5 \times 10^{-8}$). Dudbridge[46] and many others have shown, however, that such an approach excludes many useful variants, harming power/precision, and lower thresholds (e.g. $10^{-5}$) often result in improved performance. Clearly such lower threshold could result in the inclusion of (many) null-variants. However, by employing the two-sample MR paradigm (using genetic risk factor and outcome estimates from different samples), any possible weak-instrument bias will attenuate results towards the null[47].

**False positive rate.** Throughout this manuscript we use a type I error rate of 0.05 (or 95% confidence interval) and do not correct for multiple testing. While appropriate multiplicity protection is important, by focussing on four thoroughly studied drug targets (NPC1L1, HMGCR, PCSK9, and CETP) there is an abundance of prior evidence on the expected CHD effect, making analytical control of the false positive rate less relevant. In other settings, for example gene-based MR analysis of all druggable genes, appropriate control of false discovery rates is clearly essential. It could be argued that applying a genome-wide association $p$ value threshold (e.g. $5 \times 10^{-8}$) would be needlessly conservative. Instead one could control for the number of druggable proteins (about 5000; resulting in a $1 \times 10^{-5}$ threshold). However, (early) drug development is not performed in isolation, and genetic evidence will be evaluated alongside evidence from cells, tissues, and animal experiments. As such, appropriate false discovery control will depend on the position of drug target MR within this pre-existing evidence framework. A $p$ value threshold of $1 \times 10^{-5}$ might be applied when drug target MR is used as a screening tool, before validating promising leads in further experiments. Positioning drug target MR after successful in vivo experimentation, for example, to check for possible unknown side effects in human subjects, will likely call for a less-stringent multiplicity correction considering the more extensive prior knowledge and the aim of early detection of possible safety concerns. We also emphasise that the range of druggable proteins is not fixed. Indeed, our own previous paper expanded the

druggable genome from around 2000 to over 4000 proteins[11]. Moreover, new therapeutic modalities e.g. RNA silencing, are extending the range of therapeutic targets from those that are currently amenable to the action of small molecules, peptides and monoclonal antibody therapeutics that target proteins, and which remain the mainstay of drug development.

**MR estimators.** In the current manuscript we pursued drug target MR by applying a generalised least squares (GLS) solution[45], modelling residual LD, to genetic *cis*-regions known to encode protein drug targets. This GLS method is by no means the only relevant estimator function, and one can 'repurpose' many general MR methods for use in drug target MR. For example, the MR-base platform clumps variants to such a low level (e.g. R-squared of 0.001) that one can apply weighted regression solutions (e.g. IVW), foregoing LD correction. Generalised Summary-data-based Mendelian Randomisation (GSMR)[48] provides similar LD-modelling MR functionality as the GLS method applied here, which GSMR extents by allowing for automated outlier removal through HEIDI, as well as providing a solid integration with the Genome-wide Complex Trait Analysis suite. Similar outlier removal steps can be readily implemented using the Q-statistic, and standard leverage or Cook's statistics. Automated outlier removal does however make an implicit assumption that the outlying observations are incorrect, and not the statistical model; which is unlikely to be generally true. Nevertheless, outlier removal is an important step in assessing the robustness of results.

Clearly, if for any given locus one could perfectly distinguish the causal variants from null-variants, simply selecting the causal set of variants for MR will result in the most precise/powerful analysis. However, such 'oracle' selection is unlikely in practice and difficult to scale. As such the proposed LD modelling approach will not in general select the perfect (i.e. the causal) set of variants, but instead it suggests a robust set, which uses variants in LD with (unknown) causal variants as sentinels. Combining LD modelling with clumping requires limited human input and is therefore highly scalable. Finally, we note, the analyses presented in Figs. 2 and 5 are intended as an illustration of LD modelling, not as proof. The proof follows from straightforward statistical argument and simulation studies conducted by Burgess et al.[45], and the seminal work from Yang et al.[49] on COJO.

**Statistical analysis.** MR was conducted using the 'Inverse Variance Weighted' (IVW) and 'MR-Egger' methods for correlated variants as detailed in Burges et al.[18]. Here we note that these methods are specific parametrizations of GLS technique and simply refer to IVW as GLS, and MR-Egger as GLS with Egger correction.

In the context of MR, a GLS without an Egger correct, regresses the genetic association with an outcome (CHD in our case) on the genetic association with an exposure (here lipids, protein level or expression level), forcing the intercept through zero; reflecting the no-pleiotropy assumption when a zero-exposure effect should be matched by a zero outcome effect. Here the slope estimate equates to a causal estimate of the exposure to outcome effect. GLS with Egger correction refers to a similar linear model without forcing the intercept through the origin. Here the intercept estimate reflects the amount of horizontal pleiotropy, while the slope estimates reflects the causal estimate of the exposure on the outcome corrected for (potential) horizontal pleiotropy. Estimates are presented as fixed effects (with a regression standard error of unity), or as random effects (where the regression standard error is equal or larger than 1).

The drug target phenome-wide analysis was conducted by mapping the INTERVAL pQTL GWAS to the druggable genome[11] and selecting the five most significant proteins with sufficient *cis* variants to conduct further analyses. Variants were selected from a 2 kB window around the gene, excluding variants with a MAF of 0.05 or lower. The final set of instruments were selected based a LD-clumping algorithm where the LD-threshold is selected through comparison of the point estimates of threshold $l$ to $l - 1$; overinfluential (high leverage) or outlying variants were removed. Due to INTERVAL's modest sample size the number of available variants were often limited, forcing us to use the IVW estimator and forgoing any eQTL screening for possible LD-related horizontal pleiotropy.

**Reporting summary.** Further information on research design is available in the Nature Research Reporting Summary linked to this article.

## Data availability

All data are publicly available. Information on the four positive control loci (*HMGCR, PCSK9, CETP,* and *NPC1L1*) were sourced from the druggable genome (defined in ref. [11]). Specifically, for the current analyses we identified variants within a megabase upstream or downstream from each of the four loci. Outcome data were extracted from CARDIOGRAMplusC4D[50] including the genetic association (log odds ratio) with CHD, as well as their standard errors. Exposure data were leveraged from GLGC[16] (lipids), Pott et al.[28] (PCSK9 protein level), Blauw et al.[27] (CETP protein level), and GTEx[51] version 7 (expression level).

The 1000 genomes[52] data were used as a source of LD. Enhancer data were derived from the Human ACtive Enhancer to interpret Regular variants (HACER[53]) resource. All information was curated and normalised to genetic build 37 as described in detail in Finan et al.[11] A ±2.5 kB subset of the data is provided in Supplementary Tables 1–4, with the remainder easily extracted from cited publicly available sources.

The phenome-wide scan utilised INTERVAL pQTL[4] data from 5 drug targets and evaluated the protein effects on 35 outcomes using following publicly available resources: UK biobank data (nealelab.is/uk-biobank) were used for lipids (LDL-C, HDL-C, triglycerides [TG], lipoprotein A [LPa], Apolipoprotein B [ApoB], Apolipoprotein A1 [ApoA1]), glucose and HbA1c, leucocytes, lymphocytes, monocytes, and neutrophils counts. Blood pressure (systolic and diastolic [SBP, DBP]) data were used from Evangelou et al.[54], which includes the UKB as well. CKDGen consortium data provided information on blood urea nitrogen (BUN), estimated glomerular filtration rate (eGFR), and chronic kidney disease (CKD)[55]. Bone mineral density (BMD)[56] and fracture[57] data were obtained from GEFOS Consortium. Genetic associations with 'general cognitive function' were obtained from a meta-analysis of CHARGE, COGENT and UK biobank[58]. Data on CHD were available from CardiogramplusC4D[50], any stroke, large artery stroke, cardioembolic stroke, and small vessel stroke from the MEGASTROKE consortium[59], Heart Failure (HF) from the HERMES[60], atrial fibrillation (AF) from the AFgen consortium[61], and finally non-ischaemic cardiomyopathy (CM) from GRADE investigators[62]. Additional non-CVD phenotype data was extracted for type 2 diabetes (T2DM)[63], Asthma[64], inflammatory bowel disease (IBD)[65], Chron's disease (CD)[66], ulcerative colitis (UC)[67], multiple sclerosis (MS)[68] and Alzheimer's disease[69].

## Code availability

All analyses were conducted using the R programming language[70], with packages dplyr[71], ggplot2[72], gridExtra[73], openxlsx[74], and wesanderson[75]. Diagrams were programmed in TikZ[76], and the Supplementary written in LaTeX and knitr[77]. The grid search can be readily implemented with a generic statistical programming language and irrespective of the desired MR estimator.

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

## Acknowledgements

AFS is supported by BHF grant PG/18/5033837 and the UCL BHF Research Accelerator AA/18/6/34223. CF and AFS received additional support from the National Institute for Health Research University College London Hospitals Biomedical Research Centre. MGM is supported by a BHF Fellowship FS/17/70/33482. RF is supported by UK Medical Research Council (FC001002, MR/N013867/1). MZ contributed to this work as part of her PhD, which was funded by BenevolentAI, where she was an employee. ADH is an NIHR Senior Investigator. We further acknowledge support from the Rosetrees and Stoneygate Trusts. FWA is supported by UCL Hospitals NIHR Biomedical Research Centre. RSP is supported by a BHF Fellowship FS/14/76/30933. This work has received support from the EU/EFPIA Innovative Medicines Initiative [2] Joint Undertaking BigData@Heart grant n° 116074. This research has been conducted using the UK Biobank Resource under Application Number 12113. The authors are grateful to UK Biobank participants. UK Biobank was established by the Wellcome Trust medical charity, Medical Research Council, Department of Health, Scottish Government, and the Northwest Regional Development Agency.

## Author contributions

AFS, CF, MGM, RF, MZ and ADH contributed to the idea, and design of the study, AFS, CF, SC and MGM performed the analyses. AFS, CF and ADH drafted the manuscript. AFS, CF and SC curated, extracted and normalised these data. MG, RP, DFF, BT, FWA, SC, RF and MZ provided critical input on the analyses and the drafted manuscript.

## Competing interests

DFF is a full-time employee of Bayer AG, Germany. BT is a full-time employee of Servier. RSP has received honoraria from Sanofi, Bayer and Amgen. MZ is a full-time employee of GSK. AFS and FWA have received Servier funding for unrelated work. MZ conducted this research as an employee of BenevolentAI. Since completing the work MZ is now a full-time employee of GlaxoSmithKline. None of the remaining authors have a competing interest to declare.

## Additional information

**Peer review information** *Nature Communications* thanks Eric Fauman and Heiko Runz for their contribution to the peer review of this manuscript. Peer review reports are available.

