## [Peer Review File · Nature Communications]

Reviewers' comments:

Reviewer #1 (Remarks to the Author):

This paper presents a novel framework for genetically validating specific potential drug targets essentially by performing a locus-specific Mendelian Randomization analysis.

This is a nice idea and will definitely become more valuable and feasible as we accumulate the necessary data to properly perform such analyses. Principally this will be of the form of tissue-relevant protein QTL data.

Unfortunately, such data is not currently available. The authors attempt to ameliorate this in a few ways, but this introduces new issues.

On page 9, the authors suddenly switch to using LDL-C instead of protein abundance as introduced earlier in the paper. This does make some sense, but it feels a bit like a "bait-and-switch" in the overall arc of the paper as the discussion earlier is focused on protein abundance. Of course Sek Kathiresan and colleagues demonstrated the causal role of LDL-C on risk of MI via MR in their classic 2012 Lancet paper. I believe what's novel here is the validation on a locus-by-locus basis that the general LDL-C influence on MI is recapitulated at each drug target locus. I believe the 2012 Lancet paper isn't even referenced. I think it would be valuable to discuss this approach and these results in the context of the 2012 paper.

It's not clear from the paper why they selected HDL-C as the intermediate trait for CETP instead of LDL-C. CETP certainly has genome-wide significant associations with LDL-C levels. In particular, it would be interesting to see the OR for CHD per sd of LDL for CETP along side those for NPC1L1, PCSK9 and HMGCR. If they selected HDL-C as the strongest association at CETP, this should be stated, but I still believe the LDL-C analysis would be instructive.

The use of eQTLs as the exposure trait introduces multiple new issues. First is the assumption that mRNA abundance correlates with protein abundance, which is frequently not the case. See for example Chick et al, Nature 2016, Defining the consequences of genetic variation on a proteome-wide scale. The authors state an assumption that there exists "a certain proportionality between mRNA and protein expression", which I believe is unjustified as a general rule, even if it may apply to specific proteins. Clearly before applying this approach one would want to demonstrate a fixed proportionality for the specific gene of interest.

The authors make much of the fact that different tissues give different eQTL results, even differing in directionality of impact. While this is certainly true for expression data, it must also apply to protein and biomarker abundance. Currently we mostly rely circulating protein and biomarker data and this shields us from the inevitable ambiguity we'll observe when we have bulk pQTL and biomarker QTL data across multiple tissues and organs. The authors should directly address this and suggest approaches to deal with this ambiguity when it arrives.

The authors benefit from the fact that for LDL-C most of the biology is presumably occurring in the blood, but this will not be true of many or most diseases. This limitation should be acknowledged and addressed.

Overall I appreciate the contribution of this paper in developing methods to apply MR to the validation of individual potential drug targets. I think there are a number of assumptions and caveats as outlined above that should be fully discussed in the manuscript prior to publication.

Small additions/corrections/suggestions:

In the abstract:

Qualify this statement: Proteins are [often/usually/frequently/typically] the proximal effectors
Missing parenthesis: platforms, (e.g. from
Could omit the following sentence – not clear how this adds to the point of this paper:
Additional resources that can be utilised include GCTA, PrediXcan14 and MetaXcan15.

Line 81 page 4 – I think the argument can be both both strengthened and weakened.
In favor of proteins as an instrument over transcripts is the point (made earlier in the manuscript) that generally the biology (signaling, chemical reaction, transport) is carried out by the protein, not the transcript, so protein abundance is a more relevant exposure trait than mRNA abundance. A caveat to the “Central Dogma” argument is that theoretically you could still have reverse causality with the SNP affecting disease state which then drives protein abundance although this should be greatly mitigated by only considering cis-pQTLs.

Line 151 page 7 – I had to read this sentence three or four times before I felt I understood the point the authors are trying to make. I’m not sure it adds anything to the paper and should perhaps be omitted:

In the context of MR analysis of proteins, vertical and horizontal pleiotropy correspond to ‘pre-’ and ‘post’-translational effects respectively.

Line 348, page 15: Grammer:
...we accounted (conditioned) for pairwise LD used the European (EUR) 1000 genomes panel.

Line 532, page 22: slight typo:
clearly essential.it could be argued

Line 575, page 24: slight typo
level, or exposure level),
I think the authors mean “expression” level

Reviewer #2 (Remarks to the Author):

In this manuscript, the authors propose proteins as more reliable exposures for Mendelian Randomization than other biological measures. They introduce an alternative for pQTL-based cis-MR using instruments across an entire cis-region instead of distinct functional variants. At the case of four well-characterized lipid and CAD loci that encode targets for well-established drugs (HMGCR, PCSK9, NPC1L1, CETP) they introduce the concept of “drug target MR”, claiming their approach might increase precision and power of MR to validate drug targets. This is a well conducted and interesting study expanding the steadily increasing toolkit for MR. However, the mathematical concept appears to yield at best incremental insights over existing tools, and the analysis of just four drug targets that all have already been well studied earlier using MR adds little to existing knowledge.

1. The authors list as one main argument why cis-instruments should be lumped together for protein-based MR the scarcity of functional variants, and specifically pQTLs, for most drug targets. However, they do not systematically compare their method to existing data and approaches beyond just two targets, despite extensive pQTL-based MR data on drug targets at many other GWAS loci and assessed with various platforms being publicly available (e.g., from Sun et al., Ref. 8). How well does their approach compare to existing approaches across more targets? Can it live

up to its promise when conducted at a much larger scale?

2. The argument that cis-instruments may be “less prone to violating the ‘no horizontal pleiotropy’ assumption” is unfounded. Unfortunately, a large number of cis-pQTLs have been demonstrated to also be trans-pQTLs, and there are numerous published examples where variants with the 2.5kB cut-off used in the paper impact mRNA and protein levels in trans. What evidence can the authors provide that their hypothesis is correct?

3. If the MR analysis is aiming at estimating the effect of protein (P) on disease (D), the distinction between direct (ϕ_G) and indirect ($\mu\theta$) effects of the protein on disease is irrelevant. The authors simply expand the P \rightarrow D path to include X, which is ignored in the protein-disease MR analysis described in lines 181-182 (since they eventually only provided an estimator for ω = the sum of direct and indirect effects of protein on disease). The real horizontal pleiotropy that they should worry about is the existence of ϕ_G , which could exist if 1) G has cis-effects on other proteins encoded in the same region, 2) G has trans-effects on other proteins, 3) G has effects on disease through other biological pathways that does not involve protein level change. Their lumping approach does address this. Also, this paper does not contribute to assess how MR could help to determine that ϕ_G could be in fact 0 (one of the key challenges in drug target validation).

4. The authors discuss at length a smaller set of <4,500 “druggable genes”, but such dimension reduction from the full protein-coding genome is not being leveraged, e.g. for analyzing more than just the selected loci, or for multiple testing correction (the authors admit their current significance cut-offs are fairly lenient). Also, that “cis-MR for drug target validation requires the selection of genes for druggable proteins” (line 241) is plainly wrong: others have applied MR to targets that would not fall into this category (see e.g. Sun et al., Ref. 8), and the concept of what constitutes a good drug target has changed substantially over the recent years (see e.g., Plenge et al., Nat Rev Drug Disc 2013; PMID: 23868113 for discussing a more genetics perspective). The term “drug target MR” introduced here is thus certainly overstated: the method presented here might serve as one additional useful approach to validate drug targets through MR, but would most likely be used as one of several methods. I speculate that for most targets it will probably not turn out as the best one, but it’d be great to be convinced otherwise.

Reviewer 1

1) This paper presents a novel framework for genetically validating specific potential drug targets essentially by performing a locus-specific Mendelian Randomization analysis. This is a nice idea and will definitely become more valuable and feasible as we accumulate the necessary data to properly perform such analyses.

Response: We thank the reviewer for recognising the novelty and importance of the approach described in the manuscript.

2) Principally this will be of the form of tissue-relevant protein QTL data. Unfortunately, such data is not currently available. The authors attempt to ameliorate this in a few ways, but this introduces new issues.

Response: We thank the reviewer for raising this important issue. Our aim was to use biological and mathematical arguments to indicate why locus-specific rather than genome wide Mendelian randomisation (MR) analysis is the appropriate approach for genetic validation of drug targets; but also, to outline and illustrate the challenges and decisions involved in such analyses. These include defining the loci of interest; selecting and accounting for linkage disequilibrium between genetic variants at a locus; and selecting the exposure variable through which to weight the effect of the selected genetic instruments; which is relevant to the specific point raised by the reviewer.

In relation to the selection of the exposure variable, we considered the *currently available* choices, and considered the strengths and limitations of each. The available exposures for two-stage locus-specific MR analysis include: (locus- and tissue-specific) mRNA expression; the concentration of the encoded protein in the circulation, where this is found in the circulation and can be measured; or some downstream biomarker of protein action, where this is known and has been measured (for example, a circulating lipid fraction for a protein that affects lipid metabolism). The genetic variants associated with such exposures are available from GWAS data deposits or resources such as GTEX. Measures that might become available in the future are tissue- and, eventually, cell- specific protein expression.

We agree with the reviewer that 'tissue-relevant protein QTL data' may emerge in the future as the optimal exposure in locus-specific MR analysis for drug target validation. However, some of the challenges that currently apply to tissue-specific eQTL data (discussed further below) are likely shared by tissue-specific pQTLs.

We would also add two points:

First, many of the circulating proteins measured by the currently available proteomics platforms (e.g. from Somalogic [2036 druggable proteins] or O-link [973 druggable proteins]) are the *actual efficacy targets* for many approved or developmental monoclonal antibody therapeutics. Thus, the available *circulating* (rather than tissue-specific) proteomics data is already providing a step change in the ability to apply MR for genetic target validation through direct assay of the efficacy target. We have included analyses on five such examples in the revised manuscript to address this and other points raised by reviewer 2 (*vide infra*).

Second, there has been no previous systematic analysis of the tissue expression profiles of genes encoding drug targets (nor tissue specific pQTLs), so identifying the ‘relevant’ (as opposed to convenient) target tissue for a drug target MR analysis is difficult using eQTL weighted analysis and currently not possible using pQTL weighted analysis as the reviewer indicates. As part of a separate but related project, we have recently undertaken an analysis that shows that, on average, the expression of drug target genes is broader than might be suspected, but that certain drug target encoding genes exhibit very narrow and others very broad expression profiles. We will report these findings in a separate paper. Therefore, to reflect the important general point raised by the reviewer, on the optimal exposure, and the related issues above, we have amended the manuscript on page 24 as follows:

“We have used biological and mathematical arguments to formalise the distinction between locus-specific Mendelian randomisation (MR) analysis for drug target validation, where the appropriate instruments are variants in or within the vicinity of the encoding genes, and the other types of Mendelian randomisation analysis, e.g. for biomarker validation, where instruments are used from throughout the genome. Using algebraic derivations, we show that because drug target MR considers the effects of perturbing a protein drug target on disease, this type of MR may be applied in settings where biomarker MR could be biased through horizontal pleiotropy.

We also illustrate the challenges, and the choices to be made, when undertaking Mendelian randomisation for drug target validation. These include defining the loci of interest; selecting and accounting for linkage disequilibrium between genetic variants to be used as instruments; and selecting the exposure through which to weight the effect of the genetic instruments used.”

We have also added the following on pp. 28-29:

“Over 90% of drug targets are proteins, therefore weighting by protein expression in a disease-relevant tissue would provide the most informative *cis*-MR analysis for drug target validation. Many of the circulating proteins measured by the currently available proteomics platforms (e.g. from Somalogic [2036 druggable proteins] or O-link [973 druggable proteins]) are the actual biological efficacy targets of many licensed or developmental peptide or monoclonal antibody therapeutics. Thus, the available data on the circulating proteome already provides a step change in the ability to apply genetic target validation. We have illustrated the potential of this approach by conducting a *cis*-MR and PheWAS for five such targets.

Where drugs bind membrane bound or intracellular proteins which subsequently affect non-protein constituents of the circulating blood, it becomes possible to anticipate their effects using MR studies in which instruments in the gene encoding the corresponding drug target are weighted by their effect on a relevant circulating non-protein biomarker. We illustrated this with reference to four drug targets for lipid lowering (HMGCR, NPC1L1, PCSK9 and CETP). For two of these targets (PCSK9 and CETP) it was also possible to compare the findings of *cis*-MR analyses weighted by the level of the circulating protein with *cis*-MR analyses weighted by the relevant lipid fraction.

However, for many drugs or drug targets, for example, those used in, or being developed for, the treatment of neurological, myocardial or musculoskeletal conditions, a circulating biomarker may be unavailable or may not represent a strong proxy for the drug target. Tissue-specific pQTL data has yet to be generated at any scale and, until such data become available, we investigated tissue specific eQTLs as a potentially relevant alternative exposure that might closely proxy pQTL effects. However, we found that eQTL-based MR estimates may differ both in magnitude as well as direction across tissues, as demonstrated by exhaustive analyses of the *HMGCR*, *NPC1L1*, *PCSK9* and *CETP* loci.

This tissue-dependent heterogeneity was independently reported by the GTEx consortium for *PCSK9*⁶⁶. We extend those observations to demonstrate their potential to undermine reliable causal inference when using mRNA expression as a weighting variable in MR analysis. Possibly, the observed heterogeneity may relate to the inclusion of non-European ancestries in the GTEx database⁶⁷, or due to the post-mortem collection of samples⁶⁸. For example, GTEx previously reported that gene expression changed post-mortem in a tissue-specific manner, which they attempted to ameliorate with multiple regression⁶⁸.

This tissue-specific heterogeneity likely reflects actual biology which might also extend to tissue-specific pQTL data. A key uncertainty is identifying the 'relevant' tissue for a drug target validation MR analysis. This would be informed by greater knowledge of the expression profiles of all human drug targets, an area that has so far received limited attention. When these data become available it will become important to evaluate tissue dependency and the underlying mechanisms in more detail. Where it is relevant, and possible, to use circulating proteins or biomarkers such as lipids as exposure variables in two-stage drug target MR analysis this may help mitigate the complexity of weighting based on tissue-specific eQTL or pQTL data."

3) On page 9, the authors suddenly switch to using LDL-C instead of protein abundance as introduced earlier in the paper. This does make some sense, but it feels a bit like a "bait-and-switch" in the overall arc of the paper as the discussion earlier is focused on protein abundance.

Response: We apologise that the argument on the use of downstream biomarkers of protein action as an exposure variable in drug target MR analysis did not flow logically as part of the narrative arc as we intended it to. We have therefore amended the introduction as follows to rectify this issue.

On page 6, we write:

"Further questions include whether to weight such instruments in an MR analysis by the level of protein *expression* or *activity*, where the relevant assays are available; or, where they are not, by the level of mRNA expression (and, if so, in which tissue), or by some downstream consequence of protein action, e.g. differences in the level of a metabolite known to be influenced by the protein such as LDL-C for a lipid-lowering drug target."

On page 7, we now write:

"Tissue specific eQTLs have been reported for all four genes and pQTL data for two of the proteins that circulate in the plasma (*PCSK9* and *CETP*) allowing comparisons to be drawn between *cis*-MR analysis weighted by mRNA expression, protein expression and effects on downstream biomarkers, in this case the circulating lipid fractions LDL-C, HDL-C and triglycerides."

4) Of course Sek Kathiresan and colleagues demonstrated the causal role of LDL-C on risk of MI via MR in their classic 2012 Lancet paper. I believe what's novel here is the validation on a locus-by-locus basis that the general LDL-C influence on MI is recapitulated at each

drug target locus. I believe the 2012 Lancet paper isn't even referenced. I think it would be valuable to discuss this approach and these results in the context of the 2012 paper.

Response: The excellent paper of Kathiresan and colleagues and other work (including our own – Holmes EHJ 2014) has demonstrated the use of Mendelian randomisation analysis to assess the causal relevance of the major circulating lipid fractions (LDL-C, HDL-C and triglycerides) for coronary heart disease (CHD). The reviewer is correct that whereas these analyses have utilised variants from throughout the genome as instruments, the approach we describe uses genetic instruments restricted to the target of interest (acting in *cis*). The two approaches are both relevant but seek to answer different questions. Whereas the former addresses the causal relevance of the biomarker (e.g. LDL-C) for CHD, the latter seeks to address whether modification of a specified drug target (e.g. PCSK9) will reduce CHD, and is more directly relevant to drug development. In these cases, genetic effects on LDL-C (for example) are merely used as a proxy for the unmeasured genetic association with PCSK9, a protein which is known to affect LDL-C. The new cross plots introduced in the **Appendix Figure 1**, which utilise GWAS summary statistics from the Global Lipid Genetics Consortium of blood lipids and the CardiogramPlusC4D consortium analysis of CHD, contrast biomarker and drug target MR analysis pictorially. Whereas variants in the genes encoding representative drug targets for LDL-C modification show effects on CHD risk that are consistent with their association with LDL-C (and, for these licensed drug targets, with the results of clinical trials), variants in genes encoding drug targets affecting HDL-C concentration show effects on CHD risk that are not readily anticipated from their effect on HDL-C. The examples illustrate why MR analyses of a circulating biomarker using variants selected from throughout the genome might not reliably infer the effect on a disease end-point of modifying a specific target.”

To incorporate the reviewer's suggestion to discuss the approach in relation to the Kathiresan paper and blood lipids, we have amended the text on page 4 as follows:

“For example, prior work has demonstrated the use of Mendelian randomisation analysis to assess the causal relevance of the major circulating lipid fractions (LDL-C, HDL-C and triglycerides) for coronary heart disease (CHD). These analyses have utilised variants from throughout the genome as instruments, whereas the approach we explore in this paper uses genetic instruments restricted to the target of interest (acting in *cis*). The two approaches are both relevant but seek to answer different questions. Whereas the former addresses the causal relevance of the biomarker for CHD, the latter seeks to address whether modification of a specified drug target will reduce CHD, and uses the biomarker as a proxy of protein level and activity. These two approaches may yield different estimates when the protein drug target, for example CETP, affects multiple biomarkers (e.g., HDL-C, and LDL-C) through so-called post-translational pleiotropy (see below).

In the case of blood lipids, whereas previous clinical trials have indicated that LDL-C lowering by whatever means is likely to reduce CHD risk, this has not proved the case for HDL-C. The cross plots in Appendix 1, contrast biomarker and drug target MR analysis pictorially. While variants in the genes encoding representative drug targets for LDL-C modification show effects on CHD risk that are consistent with their association with LDL-C (and, for licensed drugs, the results of clinical trials), variants in genes encoding drug targets affecting HDL-C concentration show effects on CHD risk that are not readily anticipated from their effect on HDL-C. The examples illustrate why MR analyses of a circulating biomarker using variants selected from throughout the genome might not reliably infer the effect on a disease end-point of modifying a specific target.”

5) It's not clear from the paper why they selected HDL-C as the intermediate trait for CETP instead of LDL-C. CETP certainly has genome-wide significant associations with LDL-C levels. In particular, it would be interesting to see the OR for CHD per sd of LDL for CETP along side those for NPC1L1, PCSK9 and HMGCR. If they selected HDL-C as the strongest association at CETP, this should be stated, but I still believe the LDL-C analysis would be instructive.

Response: We thank the reviewer for raising this point and we address it in three ways.

The rationale for selecting HDL-C as the intermediate trait was that CETP inhibitors were developed on the basis of their HDL-C raising property (Brousseau et al, NEJM 2004), motivated by the inverse observational association between HDL-C and CHD (The Emerging Risk Factors Collaboration, JAMA 2009), and the finding that individuals with rare null mutations in CETP have very substantial elevations in HDL-C. Nevertheless, CETP inhibitor drugs also lower LDL-C and raise triglycerides, an effect shared by variants in the CETP gene. The finding that the CETP inhibitor drug anacetrapib also reduces CHD risk in a clinical trial has prompted speculation that the benefits arise the not from the HDL-C elevating but from the LDL-C lowering effect.

The reviewer's suggestion is therefore a very reasonable one and we have repeated the analyses for CETP using LDL-C weights and show consistent findings to those using HDL-C weights. We have amended the manuscript as follows on page 16:

"Given the ongoing debate on whether the beneficial effect of CETP-inhibition depends on HDL-C raising or LDL-C lowering, we repeated these analyses using LDL-C weights (Appendix 6), with similar results to those observed using HDL-C weights."

Second, we have expanded the mathematical derivation of locus-specific drug target MR analysis. If the inference to be drawn is the protein effect on a disease outcome (CHD for example), performing a drug target MR using a biomarker downstream of the protein (as in HDL-C and LDL-C in the case of CETP), does not require the downstream biomarker to be causally related to CHD, though it may well be. Instead, all that is required is for the protein to affect the downstream biomarker. Clearly, based on the drug trial results, this requirement is met for CETP and HDL-C or LDL-C.

We have added the following to the manuscript on page 11:

"Furthermore, if the protein has a direct effect on the disease, that is not mediated by the downstream biomarker, $\omega_{bw} \neq 0$ does not provide evidence for the biomarker itself to causally effect disease; i.e, $\omega_{bw} \neq 0$ does not imply $\theta \neq 0$. The only additional requirement for using a "biomarker weighted" MR for drug target validation is that the protein is strongly correlated with the downstream biomarker, for example when $\mu \neq 0$; which is a (slightly) different version of IV assumption (i)."

Third, we incorporate a *cis*-MR analysis for CETP weighted by circulating CETP concentration (a *cis*-pQTL analysis) rather than any specific blood lipid fraction. We have added the following on p 18:

“Of the four proteins considered here we had access to pQTL estimates from two GWAS’ of circulating CETP and PCSK9 concentration measured by enzyme-linked immunosorbent assays (ELISA)^{44,45}, measured in 4000 and 3000 subjects, respectively. Initially focussing on variants in the same ± 2.5 KB region as before, we found circulating CETP increased CHD risk, consistent with the findings of a recent large-scale clinical trial where CETP inhibition reduces CHD risk (Figure 5). Similarly convincing results were seen for the analyses of log(PCSK9) concentration and CHD”

6) The use of eQTLs as the exposure trait introduces multiple new issues. First is the assumption that mRNA abundance correlates with protein abundance, which is frequently not the case. See for example Chick et al, Nature 2016, Defining the consequences of genetic variation on a proteome-wide scale. The authors state an assumption that there exists “a certain proportionality between mRNA and protein expression”, which I believe is unjustified as a general rule, even if it may apply to specific proteins. Clearly before applying this approach one would want to demonstrate a fixed proportionality for the specific gene of interest.

Response: We agree entirely that the use of eQTLs introduce new issues that we sought to highlight and explore as an important component of the manuscript. We felt this to be important as eQTLs are becoming widely used in such analyses motivated by the development of tools such as GSMR and S-Predixscan.

We noted that the use of eQTLs requires the assumption of a proportionality between mRNA and protein expression but, as the reviewer points out, we did not give sufficient discussion to how likely this is to be the case. Chick et al. compared e- and pQTLs in liver tissue from 192 outbred mice and found that 1400 of 6707 proteins studied had overlapping e and pQTLs and that among these the correlation between mRNA and protein abundance has a mean correlation of 0.5. Among those proteins without a shared e and pQTL, the mean correlation was around 0.25.

We have therefore amended the manuscript as follows on pages 19:

“Because tissue-specific eQTL data is more widely available than tissue-specific pQTL data, we additionally evaluated the performance of MR analysis using mRNA expression level as the exposure variable. An analysis of this type assumes a certain proportionality between mRNA and protein expression, but the strength of this assumption has not been tested systematically across human tissues. We note that Chick et al⁴⁶. compared eQTLs and pQTLs in liver tissue from 192 outbred mice and found that 1400 of 6707 proteins studied had overlapping eQTLs and pQTLs, and among these overlapping proteins showed a high. Among proteins without shared eQTLs and pQTLs, the mean correlation was around 0.25. Recognising this caveat, we obtained information on genetic effects on mRNA expression from GTEx version 7, for (1MB) *cis*-regions of all four genes studied here, based on post mortem tissues from 449 donors (84% of European descent).”

7) The authors make much of the fact that different tissues give different eQTL results, even differing in directionality of impact. While this is certainly true for expression data, it must

also apply to protein and biomarker abundance. Currently we mostly rely circulating protein and biomarker data and this shields us from the inevitable ambiguity we'll observe when we have bulk pQTL and biomarker QTL data across multiple tissues and organs. The authors should directly address this and suggest approaches to deal with this ambiguity when it arrives.

Response: Thanks, we have added the following statement to the manuscript on page 30.

“This tissue-specific heterogeneity likely reflects actual biology which might also extend to tissue-specific pQTL data. A key uncertainty is identifying the ‘relevant’ tissue for a drug target validation MR analysis. This would be informed by greater knowledge of the expression profiles of all human drug targets, an area that has so far received limited attention. When these data become available it will become important to evaluate tissue dependency and the underlying mechanisms in more detail. Where it is relevant, and possible, to use circulating proteins or biomarkers such as lipids as exposure variables in two-stage drug target MR analysis this may help mitigate the complexity of weighting based on tissue-specific eQTL or pQTL data.”

8) The authors benefit from the fact that for LDL-C most of the biology is presumably occurring in the blood, but this will not be true of many or most diseases. This limitation should be acknowledged and addressed.

Response: Thanks. As a clarification, circulating LDL-C concentration (similar to other lipids) reflects enzymatic reactions and *transporter* functions that occur in the blood but also the gut wall (related to cholesterol absorption), in the liver (related to cholesterol synthesis) and in all cells (where cholesterol uptake and metabolism are key to normal function).

However, noting the reviewer's important point and to demonstrate generalisability to other non-lipid factors, we have now added a phenome-wide scan using pQTL protein exposures which further shows the generalizability of our approach, see pages 22-24.

Moreover, we have also now strengthened discussion of the approach to the situation where no circulating protein or non-protein biomarker is available on p.28-29 as follows:

“Over 90% of drug targets are proteins, therefore weighting by protein expression in a disease-relevant tissue would provide the most informative *cis*-MR analysis for drug target validation. Many of the circulating proteins measured by the currently available proteomics platforms (e.g. from Somalogic [2036 druggable proteins] or O-link [973 druggable proteins]) are the actual biological efficacy targets of many licensed or developmental peptide or monoclonal antibody therapeutics. Thus, the available data on the circulating proteome already provides a step change in the ability to apply genetic target validation. We have illustrated the potential of this approach by conducting a *cis*-MR and PheWAS for five such targets.

Where drugs bind membrane bound or intracellular proteins which subsequently affect non-protein constituents of the circulating blood, it becomes possible to anticipate their effects using MR studies in which instruments in the gene encoding the corresponding drug target are weighted by their effect on a relevant circulating non-protein biomarker. We illustrated this with reference to four drug targets for lipid lowering (HMGCR, NPC1L1, PCSK9 and CETP). For two of these targets (PCSK9 and CETP) it

was also possible to compare the findings of *cis*-MR analyses weighted by the level of the circulating protein with *cis*-MR analyses weighted by the relevant lipid fraction.

However, for many drugs or drug targets, for example, those used in, or being developed for, the treatment of neurological, myocardial or musculoskeletal conditions, a circulating biomarker may be unavailable or may not represent a strong proxy for the drug target. Tissue-specific pQTL data has yet to be generated at any scale and, until such data become available, we investigated tissue specific eQTLs as a potentially relevant alternative exposure that might closely proxy pQTL effects. However, we found that eQTL-based MR estimates may differ both in magnitude as well as direction across tissues, as demonstrated by exhaustive analyses of the *HMGCR*, *NPC1L1*, *PCSK9* and *CETP* loci. This tissue-dependent heterogeneity was independently reported by the GTEx consortium for *PCSK9*⁶⁶. We extend those observations to demonstrate their potential to undermine reliable causal inference when using mRNA expression as a weighting variable in MR analysis. Possibly, the observed heterogeneity may relate to the inclusion of non-European ancestries in the GTEx database⁶⁷, or due to the post-mortem collection of samples⁶⁸. For example, GTEx previously reported that gene expression changed post-mortem in a tissue-specific manner, which they attempted to ameliorate with multiple regression⁶⁸.

This tissue-specific heterogeneity likely reflects actual biology which might also extend to tissue-specific pQTL data. A key uncertainty is identifying the 'relevant' tissue for a drug target validation MR analysis. This would be informed by greater knowledge of the expression profiles of all human drug targets, an area that has so far received limited attention. When these data become available it will become important to evaluate tissue dependency and the underlying mechanisms in more detail. Where it is relevant, and possible, to use circulating proteins or biomarkers such as lipids as exposure variables in two-stage drug target MR analysis this may help mitigate the complexity of weighting based on tissue-specific eQTL or pQTL data."

9) Overall I appreciate the contribution of this paper in developing methods to apply MR to the validation of individual potential drug targets. I think there are a number of assumptions and caveats as outlined above that should be fully discussed in the manuscript prior to publication.

Response: We thank the reviewer and hope the changes we have now made fully address the carefully highlighted issues.

Small additions/corrections/suggestions:

10) In the abstract:

Qualify this statement: Proteins are [often/usually/frequently/typically] the proximal effectors

Missing parenthesis: platforms, (e.g. from

Could omit the following sentence – not clear how this adds to the point of this paper:

Additional resources that can be utilised include GCTA, PrediXcan14 and MetaXcan15.

Response: Thanks. Amended to:

“Proteins are typically the proximal effectors of biological processes encoded in the genome, and are becoming assayable on an -omics scale.”

11) Line 81 page 4 – I think the argument can be both both strengthened and weakened. In favor of proteins as an instrument over transcripts is the point (made earlier in the manuscript) that generally the biology (signaling, chemical reaction, transport) is carried out by the protein, not the transcript, so protein abundance is a more relevant exposure trait than mRNA abundance.

A caveat to the “Central Dogma” argument is that theoretically you could still have reverse causality with the SNP affecting disease state which then drives protein abundance although this should be greatly mitigated by only considering *cis*-pQTLs.

Response: We thank the reviewer for raising this important point. In response, we have added the following on page 5:

“Finally, *cis*-MR of a protein risk factor greatly reduces the risk of reverse causation, because Crick’s dogma indicates that the pathway gene → encoded protein → disease would always be favoured over the pathway gene → disease → encoded protein, especially given that the gene → encoded protein association is typically derived from population-based (disease-free) samples. Thus, from an MR perspective, proteins are in a privileged position compared to other categories of risk factor and the use of *cis*-MR represents an optimal approach to instrument their causal effect for disease (See Appendix pages 3-5 and Appendix Figure 2-3)”

12) Line 151 page 7 – I had to read this sentence three or four times before I felt I understood the point the authors are trying to make. I’m not sure it adds anything to the paper and should perhaps be omitted:

In the context of MR analysis of proteins, vertical and horizontal pleiotropy correspond to ‘pre-’ and ‘post’-translational effects respectively.

Response: When proteins serve as the exposure of interest in Mendelian randomisation analysis it becomes possible to give biological context to the somewhat abstract concepts of vertical and horizontal pleiotropy. This is because horizontal pleiotropy equates to pathways from gene to disease which precede translation of the protein of interest, e.g. through alternative splicing or micro RNA effects. By contrast, vertical pleiotropy refers to the downstream actions of the translated protein, which should be reproduced by a drug with specific action on the protein. For this reason, we would like to retain the idea but agree we could explain it more clearly. We have changed the sentence on page 8 as follows:

“When proteins serve as the exposure of interest in Mendelian randomisation analysis it becomes possible to give biological context to the concepts of vertical and horizontal pleiotropy. This is because horizontal pleiotropy equates to pathways from gene to disease which precede translation of the protein of interest, e.g. through alternative splicing or micro RNA effects. By contrast, vertical pleiotropy refers to the downstream actions of the translated protein, which should be reproduced by

a drug with specific action on the protein. Therefore, in the context of MR analysis of proteins, vertical and horizontal pleiotropy correspond to ‘pre-’ and ‘post’-translational effects respectively”

13) Line 348, page 15: Grammer:

...we accounted (conditioned) for pairwise LD used the European (EUR) 1000 genomes panel.

Response: Thanks.

14) Line 532, page 22: slight typo:

clearly essential.it could be argued

Response: Thanks. Changed to ‘...clearly essential. It could be argued...’

15) Line 575, page 24: slight typo

level, or exposure level),

I think the authors mean “expression” level

Response: Thanks. ‘Exposure’ now changed to ‘expression’.

Reviewer 2

15) In this manuscript, the authors propose proteins as more reliable exposures for Mendelian Randomization than other biological measures. They introduce an alternative for pQTL-based *cis*-MR using instruments across an entire *cis*-region instead of distinct functional variants. At the case of four well-characterized lipid and CAD loci that encode targets for well-established drugs (HMGCR, PCSK9, NPC1L1, CETP) they introduce the concept of “drug target MR”, claiming their approach might increase precision and power of MR to validate drug targets. This is a well conducted and interesting study expanding the steadily increasing toolkit for MR. However, the mathematical concept appears to yield at best incremental insights over existing tools, and the analysis of just four drug targets that all have already been well studied earlier using MR adds little to existing knowledge.

Response: To clarify, we propose that if the aim of an MR analysis is to validate a drug target, over 90% of which are proteins, that the optimal design is a *cis*-MR analysis which utilises variants within or in the vicinity of a gene encoding the target of interest, that influence the expression or function of the encoded protein. We provide mathematical and biological arguments to support the proposal that *cis*-MR analysis is less prone to the ‘no horizontal pleiotropy’ assumption that is critical to robust causal inference.

We already carefully highlight the insights provided by the mathematical concept as follows:

- On p.10: “where the causal effect of the protein is the parameter of interest, we only need to assume that there is no direct effect of the genetic variant on disease, i.e. $\phi_G = 0$, and the protein can have any mixture of direct (ϕ_P), and indirect ($\mu\theta$) effects. For this reason, MR analysis of protein-disease relationships is less prone to violation of the ‘no horizontal pleiotropy’ assumption than MR analysis of downstream exposures”
- On p.10-11: “..a protein can remain the inferential target in an MR analysis even if it is not measured directly. ..

$$\omega_{bw} = \frac{\tilde{\delta}(\phi_P + \mu\theta)}{\tilde{\delta}\mu} = \frac{\phi_P + \mu\theta}{\mu},$$
$$= \frac{1}{\mu} \times \omega,$$

with *bw* indicating ‘biomarker weighted’. Clearly because the denominator contains $\tilde{\delta}\mu$, instead of $\tilde{\delta}$, ω_{bw} does not equal ω . However, ω_{bw} may still provide a valid null-hypothesis test of $\omega = 0$, because $\omega_{bw} \neq 0$ implies $\omega \neq 0$. Furthermore, if the protein has a direct effect on the disease, that is not mediated by the downstream biomarker, $\omega_{bw} \neq 0$ does not provide evidence for the biomarker itself to causally effect disease; i.e, $\omega_{bw} \neq 0$ does not imply $\theta \neq 0$. The only additional requirement for using a “biomarker weighted” MR for drug target validation is that the protein is strongly correlated with the downstream biomarker, for example when $\mu \neq 0$; which is a (slightly) different version of IV assumption (i)..”

- “On p.11 again: “In the absence of available measures of the protein of interest, a similar argument can be made for using mRNA expression (this time as an upstream variable) that proxies the effect of genetic variation on the level of the encoded protein again within the framework of a *cis*-MR analysis (see Appendix Figure 2):

$$\begin{aligned}\omega_{ew} &= \frac{\tilde{\delta}_{GE}\tilde{\delta}_{EP}(\phi_P + \mu\theta)}{\tilde{\delta}_{GE}} = \tilde{\delta}_{EP}(\phi_P + \mu\theta), \\ &= \tilde{\delta}_{EP} \times \omega\end{aligned}$$

Here the weighting is done by the association with of mRNA expression, and the $\tilde{\delta}$ effect has been decomposed into the variant effect on expression $\tilde{\delta}_{GE}$ and the expression effect on protein level $\tilde{\delta}_{EP}$. Similar as for ω_{bw} , the expression weighted (“ew”) drug target effect provides a valid test of $\omega = 0$ conditional on the absence of any horizontal pleiotropy predicting the protein effect; that is, a necessary assumption $\phi_G = \phi_E = 0$ (with index E for expression). It should be noted that, similar as for protein expression, mRNA expression level (eQTL) is tissue-specific, and utilizing eQTLs for drug target MRs necessitates a decision on the tissue(s) relevant for (de novo) drug development. We return to these issues later in the manuscript.”

We therefore propose that the mathematical concept we introduce *does* provide substantial, complementary insight over existing tools for MR analysis, which have largely been devoted to approaches for dealing with horizontal pleiotropy that arises when instruments for MR analysis are selected from *throughout the genome*. Such a genome-wide approach is appropriate when interest is in the causal relevance of a non-protein biomarker (e.g. a circulating lipid fraction such as LDL-C, HDL-C or triglycerides) or an environmental exposure (e.g. smoking behaviour), because there is no such thing as a *cis*-instrument for traits such as these, but likewise the inferences drawn from such an analysis relate to the biomarker or environmental exposure not to any particular drug target. As we discuss in response to Reviewer 1 and illustrate through Appendix Figures 1, MR analyses of a circulating biomarker (e.g. HDL-C) using variants selected from throughout the genome might not reliably infer the effect on a disease end-point of modifying a specific target (e.g. CETP).

On the point about generalisability, we initially focused on four well established positive controls of drug targets that are known to affect CHD and provide justification for this as follows on p.7:

‘... we select examples where the effect of a drug has already been reliably quantified on the protein of interest; on a widely measured downstream mediator of its effect; and on the disease outcome for which the treatment is indicated; and where variants in the gene encoding the drug target have been associated with effects that are consistent with this knowledge. Four genes that fulfil these criteria are *HMGCR*, *PCSK9*, *NPC1L1* and *CETP* that encode the targets of licensed or clinical phase drugs with known effects on lipids and coronary heart disease risk. Tissue specific eQTLs have been reported for all four genes and pQTL data for two of the proteins that circulate in the plasma (*PCSK9* and *CETP*) allowing comparisons to be drawn between *cis*-MR analysis weighted by mRNA expression, protein expression and effects on downstream biomarkers, in this case the circulating lipid fractions LDL-C, HDL-C and triglycerides.’

However, given the important point about generalisability and scalability raised by the reviewer, we have now undertaken new analyses on five circulating, druggable targets for which genetic instruments acting in *cis* have been identified from a GWAS of approximately 3000 circulating proteins measured in the INTERVAL study. We have used these instruments to conduct a phenome-wide scan across 35 outcomes relevant for drug development.

The following has therefore been added on pp.22-24:

“To further showcase the generalizability of drug target MR across targets and outcomes, we conducted analyses that focused on circulating proteins that are the direct efficacy targets of clinical phase developmental or licensed drugs. Genetic instruments for *cis*-MR analyses were identified from the INTERVAL study which conducted a GWAS of around 3000 circulating proteins⁸, measured using the Somalogic aptamer based proteomics platform. From the reported data we identified five proteins encoded in the druggable genome²⁷ (*F10*, *IL-12B*, *PLG*, *IL-1R1*, *MMP-9*) for which sufficient sentinel variants could be identified for the encoded protein. We then utilised the *cis*-MR approach described here to conduct phenome-wide association analysis against 35 clinically relevant disease and biomarker phenotypes (see methods).

Circulating factor X (encoded by *FX*) was associated with a higher risk of any stroke (OR 1.13 95%CI, 1.05,1.21), which is keeping with the use of direct acting anticoagulant drugs that inhibit factor X (e.g. apixaban) to prevent stroke in patients with atrial fibrillation⁴⁸. Furthermore, we found a possible effect of factor X on asthma (OR 0.78 95%CI 0.62, 0.99) (Figure 26). The monoclonal antibody ustekinumab directed against a common subunit of interleukin 12 and interleukin 23 interferes with the binding of these cytokines with the IL-12 receptor to inhibit inflammatory signalling⁴⁹. Ustekinumab has European marketing authorisation for the treatment of psoriasis and Crohn’s disease (CD) after demonstrating efficacy in clinical trials⁵⁰, and is under evaluation for ulcerative colitis (UC). Consistent with this, genetically instrumented higher interleukin-12 subunit beta (encoded by *IL12B*) was associated with a higher risk of CD (OR 2.02; 95% CI 1.48, 2.76), UC (OR 1.56; 95% CI, 1.31, 1.87), and inflammatory bowel disease (IBD) (OR 1.56 95%CI 1.31, 1.87) (Figure 26). Genetically-instrumented higher circulating concentration of plasminogen (encoded by *PLG*) was associated with a lower ischaemic stroke risk (OR 0.85; 95% CI 0.72, 1.00) (Figure 26), consistent with the known efficacy of recombinant tissue plasminogen activator (tPA) for acute ischaemic stroke⁵¹. The *PLG* association with an increased risk of any stroke (OR 1.06 95%CI 1.01, 1.11) is presumably due to an increase in haemorrhagic events. Increased levels of *PLG* were furthermore associated with increased risk of atrial fibrillation (AF), IBD, CD, Alzheimer’s disease (Figure 26) as well as lipids, and increased SBP (Figure 27) but these effects may not be observed therapeutically because tPA is given as a single dose in acute MI and ischaemic stroke. Higher circulating concentration of interleukin-1 receptor 1 (encoded by *IL1R1*) was associated with a lower risk of both CD, IBD and UC. This would be in keeping with the circulating form of the receptor functioning as decoy to reduce signalling through the membrane-bound form of the receptor by the pro-inflammatory cytokine interleukin 1 β . We also found evidence through *cis*-MR of a causative role for *MMP9* in CD and IBD. Recent phase 2 trials failed to demonstrate efficacy of anecaliximab, a monoclonal antibody targeting *MMP9* in either UC or CD^{52,53} (Figure 26). However, given the evidence from the MR analysis, further consideration should be given to the type, dose, frequency and duration of ant-*MMP9* therapy in Crohn’s disease, before this target is discounted.”

and on page 32:

“The phenome-wide scan utilized INTERVAL pQTL data from 5 drug targets and evaluated the protein effects on 35 outcomes using following publicly available resources: UK biobank data (nealelab.is/uk-biobank) were used for lipids (LDL-C, HDL-C, triglycerides [TG], lipoprotein A [LpA], Apolipoprotein B [ApoB], Apolipoprotein A1 [ApoA1]), glucose and HbA1c, leukocytes, lymphocytes, monocytes, and neutrophils counts. Blood pressure (systolic and diastolic [SBP, DBP]) data were used from Evangelou et al⁷⁰ which includes the UKB as well. CKDGen consortium data provided information on blood ureum nitrogen (BUN), estimated glomerular filtration rate (eGFR), and chronic kidney disease (CKD)⁷¹. Bone mineral density (BMD)⁷² and fracture⁷³ data were obtained from GEFOS Consortium. Genetic associations with “general cognitive function” were obtained from a meta-analysis of CHARGE, COGENT and UK biobank⁷⁴. Data on CHD were available from CardiogramplusC4D⁷⁵, any stroke, large artery stroke, cardioembolic stroke, and small vessel stroke

from the MEGASTROKE consortium⁷⁶, Heart Failure (HF) from the HERMES⁷⁷, atrial fibrillation (AF) from the AFgen consortium⁷⁸, and finally non-ischemic cardiomyopathy (CM) from GRADE investigators⁷⁹. Additional non-CVD phenotype data was extracted for type 2 diabetes (T2DM)⁸⁰, Asthma⁸¹, inflammatory bowel disease (IBD)⁸², Chron's disease (CD)⁸³, ulcerative colitis (UC)⁸⁴, multiple sclerosis (MS)⁸⁵, and Alzheimer's disease⁸⁶.

and page 34:

“The drug target phenome-wide analysis was conducted by mapping the INTERVAL pQTL GWAS to the druggable genome²⁷ and selecting the 5 most significant proteins with sufficient *cis* variants to conduct further analyses. Variants were selected from a 2kb window around the gene, excluding variants with a MAF of 0.05 or lower. The final set of instruments were selected based a LD-clumping algorithm where the LD-threshold is selected to prevent multi-collinearity issues (through comparison of the point estimates of threshold l to $l - 1$). Over-influential (high-leverage) or outlying variants were removed. Due to INTERVAL's modest sample size the number of available variants were often limited, forcing us to use the IVW estimator and forgoing any possible eQTL screening on possible horizontal pleiotropy “

We have also added Appendix Figures 26-27.

16) The authors list as one main argument why *cis*-instruments should be lumped together for protein-based MR the scarcity of functional variants, and specifically pQTLs, for most drug targets. However, they do not systematically compare their method to existing data and approaches beyond just two targets, despite extensive pQTL-based MR data on drug targets at many other GWAS loci and assessed with various platforms being publicly available (e.g., from Sun et al., Ref. 8). How well does their approach compare to existing approaches across more targets? Can it live up to its promise when conducted at a much larger scale?

Response: We apologize if the writing was not clear but do not wish to give the impression either that variants to be used as *cis*-instruments are scarce or that they should be 'lumped together'. First, we sought to distinguish *cis*-MR analysis from the approach that has become the norm in MR analysis of non-protein biomarkers or environmental exposures where a single variant is selected from each contributing locus from throughout the genome.

For this reason, we wrote on p.6:

“Since *cis*-MR analysis restricts selection of genetic instruments to those located in, or in the vicinity of the encoding gene, new questions emerge as how to optimise the selection of such variants. These include how best to select and define the loci of interest, the physical distance around each gene from which instruments might be drawn; how to select genetic variants as instruments with options including “no selection”, “selection by strength of association”, or “according to functional annotation”.

We extend the discussion of this issue on p.13 as follows (now adding some small edits as follows):

“Drug target MR focuses on a single gene known to encode a protein, and variants within and around such a gene are used to characterize the effect of the drug target on a single or multiple outcome(s)... However, typically the nature and number of causal variant(s) is (are) unknown, imposing the need for instrument selection.”

Next, since there are typically many *cis*-acting variants at a locus that could be non-coding and affect mRNA and protein expression, or be coding and affect protein function in the absence of an effect on expression we considered the effect on MR estimates of sampling variants at random, of selecting variants based on Ensembl Variant Effect Prediction that reports the functional consequence of each variant and of harnessing LD in the region to maximise information on genetic variation at a locus without including highly correlated variants that might destabilise a statistical model.

Stephen Burges (GenEpi) 2017 used simulated data to illustrate that modelling this LD increases power over simply selecting the most strongly associated variants and removing correlated variants. LD modelling increases power by forgoing the need to perfectly identify the causal variants within a specific region. Similar results were reported for the GSMR application and LD modelling. LD modelling has been applied to increase power/precision in other settings as well, for example in COJO and similar applications.

As described above, during revision we have extended the approach to incorporate data from Sun et al. on five circulating proteins that are direct efficacy targets of licensed or developmental drugs, incorporating a phenome-wide scan against 35 outcomes further illustrating the scalability and generalisability of the approach. The relevant additions are on p.22:

“Further Examples: phenome-wide drug target analysis

To further showcase the generalizability of drug target MR across targets and outcomes, we conducted analyses that focused on circulating proteins that are the direct efficacy targets of clinical phase developmental or licensed drugs. Genetic instruments for *cis*-MR analyses were identified from the INTERVAL study which conducted a GWAS of around 3000 circulating proteins⁸, measured using the Somalogic aptamer based proteomics platform. From the reported data we identified five proteins encoded in the druggable genome²⁷ (*F10*, *IL-12B*, *PLG*, *IL-1R1*, *MMP-9*) for which sufficient sentinel variants could be identified for the encoded protein. We then utilised the *cis*-MR approach described here to conduct phenome-wide association analysis against 35 clinically relevant disease and biomarker phenotypes (see methods).

Circulating factor X (encoded by *FX*) was associated with a higher risk of any stroke (OR 1.13 95%CI, 1.05, 1.21), which is keeping with the use of direct acting anticoagulant drugs that inhibit factor X (e.g. apixaban) to prevent stroke in patients with atrial fibrillation⁴⁸. Furthermore, we found a possible effect of factor X on asthma (OR 0.78 95%CI 0.62, 0.99) (Figure 26). The monoclonal antibody ustekinumab directed against a common subunit of interleukin 12 and interleukin 23 interferes with the binding of these cytokines with the IL-12 receptor to inhibit inflammatory signalling⁴⁹. Ustekinumab has European marketing authorisation for the treatment of psoriasis and Crohn’s disease (CD) after demonstrating efficacy in clinical trials⁵⁰, and is under evaluation for ulcerative colitis (UC). Consistent with this, genetically instrumented higher interleukin-12 subunit beta (encoded by *IL12B*) was associated with a higher risk of CD (OR 2.02; 95% CI 1.48, 2.76), UC (OR 1.56; 95% CI, 1.31, 1.87), and inflammatory bowel disease (IBD) (OR 1.56 95%CI 1.31, 1.87) (Figure 26). Genetically-instrumented higher circulating concentration of plasminogen (encoded by *PLG*) was associated with a lower ischaemic stroke risk (OR 0.85; 95% CI 0.72, 1.00) (Figure 26), consistent with the known efficacy of recombinant tissue plasminogen activator (tPA) for acute ischaemic stroke⁵¹. The PLG association with an increased risk of any stroke (OR 1.06 95%CI 1.01, 1.11) is presumably due to an increase in haemorrhagic events. Increased levels of PLG were furthermore associated with increased risk of atrial fibrillation (AF), IBD, CD, Alzheimer’s disease (Figure 26) as well as lipids, and

increased SBP (Figure 27) but these effects may not be observed therapeutically because tPA is given as a single dose in acute MI and ischaemic stroke. Higher circulating concentration of interleukin-1 receptor 1 (encoded by *IL1R1*) was associated with a lower risk of both CD, IBD and UC. This would be in keeping with the circulating form of the receptor functioning as decoy to reduce signalling through the membrane-bound form of the receptor by the pro-inflammatory cytokine interleukin 1 β . We also found evidence through *cis*-MR of a causative role for MMP9 in CD and IBD. Recent phase 2 trials failed to demonstrate efficacy of andecaliximab, a monoclonal antibody targeting MMP9 in either UC or CD^{52,53} (Figure 26). However, given the evidence from the MR analysis, further consideration should be given to the type, dose, frequency and duration of ant-MMP9 therapy in Crohn's disease, before this target is discounted."

Finally, the following was added to discuss these important considerations raised by the reviewer on p.26:

"Clearly, if for any given locus one could perfectly distinguish the causal variants from null-variants, simply selecting the causal set of variants for MR will result in the most precise/powerful analysis. However, such "oracle" selection is unlikely in practice and difficult to scale. As such the proposed LD modelling approach will not in general select the perfect (i.e., the causal) set of variants, but instead it will select a robust set, which uses variants in LD with (unknown) causal variants as sentinels. Combining LD modelling with clumping requires no human input and is therefore highly scalable. Finally, we note, the analyses presented in Figure 2 and 5 are intended as an illustration of LD modelling, not as proof. The proof follows from straightforward statistical argument and simulation studies conducted by Burgess et al⁵⁸, and the seminal work from Yang et al⁶¹ on COJO. "

18) The argument that *cis*-instruments may be "less prone to violating the 'no horizontal pleiotropy' assumption" is unfounded. Unfortunately, a large number of *cis*-pQTLs have been demonstrated to also be *trans*-pQTLs, and there are numerous published examples where variants with the 2.5kB cut-off used in the paper impact mRNA and protein levels in *trans*. What evidence can the authors provide that their hypothesis is correct?

Response: We respectfully disagree with the reviewer that the statement about *cis* instruments (*for MR analysis of proteins*) being less prone to violation of the horizontal pleiotropy assumption is unfounded. To be clear, the statement extracted by the reviewer comes from the following section of the introduction on p.5, which provides the context.

"...Second, aside from mRNA expression, differences in protein expression or function are the most proximal consequence of natural genetic variation. This has two consequences: frequently, variants located in and around the encoding gene can be identified with a very substantial effect on protein expression in comparison to other traits; moreover such instruments may also be less prone to violating the 'no horizontal pleiotropy' assumption' than variants located elsewhere in the genome (discussed below and ref ¹⁵). Third, in the case of MR analysis of proteins, Crick's 'Central Dogma'¹⁶ imposes an order on the direction of information flow from gene to mRNA to encoded protein, which does not extend beyond this to other biological traits that lie more distally in the causal chain that connects genetic variation to disease risk."

Thus for the avoidance of doubt, the statement about being less prone to horizontal pleiotropy relates *both* to the use of *cis instruments* and *proteins as the exposure of interest*.

The arguments that such an analysis is less prone to horizontal pleiotropy bias are based both on first-principles and on evidence.

From first principles

As we state on p.7

“For the effect estimate in MR to equate to a causal estimate the following critical assumptions should hold: (i) the genetic instrument is (strongly) associated with the exposure, (ii) the genetic instrument is independent of observed and unobserved confounders of the exposure-outcome association (which is secure because genetic variants are fixed and allocated at random), and (iii) conditional on the exposure and confounders, the genetic instrument is independent of the outcome (i.e. there is no instrument – outcome effect other than through the exposure of interest – the “no horizontal pleiotropy” assumption).”

As we write on p.8:

“The no horizontal pleiotropy assumption is violated when there are additional pathways by which the instrument may be related to the disease, sidestepping the exposure of interest. This could occur, for example if a genetic variant is in linkage disequilibrium (LD) with another variant that influences disease through a pathway distinct from the exposure, or if a genetic instrument also influences disease risk through another risk factor, located proximal to the risk factor of interest in the causal chain from gene to disease. In contrast, the association of a genetic instrument with exposures that lie in the causal chain distal to the exposure of interest (vertical pleiotropy) does not violate the assumptions underpinning MR analysis.”

We illustrate the concept of horizontal pleiotropy in manuscript Figure 1 (included here as well). Were the protein level MR is biased by the direct arrow from $G \rightarrow D$ labels ϕ_G . The potential for horizontal pleiotropy increases the more distal the exposure of interest from the genetic variants that are used as instruments, for example an MR analyses geared towards obtaining a causal effect the downstream biomarker $X \rightarrow D$ would be biased if the horizontal pathways ϕ_G and ϕ_P were none-zero.

We next illustrate the reason that *cis*-instruments are relatively protected from horizontal pleiotropy vs. *trans*-instruments in an MR analysis of protein exposures using the series of diagrams (a-c).

Panel **a** illustrates a *cis*-MR analysis where the causal effect of the protein of interest (P_1) for a disease outcome is instrumented using SNPs in its encoding gene (G_1). Proteins P_2 and P_3 are also altered by P_1 and hence are also associated with G_1 (an example of vertical pleiotropy), with P_2 being a mediator of the effect of P_1 on the disease outcome and P_3 being a bystander in this example. Instruments in G_1 are *cis*- for P_1 but *trans*- for P_2 and P_3 , but this is immaterial in this case because the instrumented protein is P_1 . The example illustrates how valid instruments for *cis*-MR can also have, and indeed would oftentimes be expected to also have, *trans*- effects. However, this does not compromise the validity of *cis* instruments because the protein instrumented (P_1) must be upstream of all of the other proteins in the causal pathway.

Panel **b** illustrates a situation where P_1 is again the protein of interest and it remains causal for a disease outcome. However, in this analysis, the effect of P_1 is instrumented by SNPs in a different gene (G_4) encoding an unmeasured protein P_4 (i.e. a *trans*-MR analysis). Here P_4 affects P_1 via a receptor and signalling cascade. In this example, P_4 has no independent effect on disease outcome: its effect is *through* the protein of interest, P_1 . For this reason, SNPs in G_4 also associate with P_2 and P_3 (*vertical pleiotropy*). In this example, a *trans*-MR analysis provides the correct inference on the causal relevance of P_1 for the disease outcome.

However, panel **c** illustrates the counterfactual situation where P_1 remains the protein of interest but this time it is *not causal* for the disease outcome. Its effect is again instrumented by SNPs in a different gene (G_4) encoding an unmeasured protein P_4 (i.e. the same *trans*-

MR analysis). However, in this scenario, P_4 affects disease outcome through a pathway independent of P_1 . This time the association of SNPs in G_4 with P_1 (and P_2 and P_3) is due to *horizontal pleiotropy* and any inference that there is a causal association of P_1 with the disease outcome is erroneous. Notably, the situation in panel **b** (where P_1 is causal) and panel **c** (where it is not) are indistinguishable using *trans*-MR analysis: in both cases G_4 associates with P_1 (the protein of interest), P_2 , P_3 and the disease outcome.

These first principles arguments, illustrated through panels **a-c**, show why the assertion is well founded that using *cis* instruments for MR analysis of proteins is less prone to violation of the horizontal pleiotropy assumption.

Empirical evidence

The reviewer also asks for empirical evidence for this assertion, which we provide using the example of the circulating protein C-reactive protein (CRP). Circulating CRP concentration is associated with later risk of CHD in non-genetic observational studies but it was previously unclear if this association is causal, confounded or arises due to reverse causation. In 2011, in a paradigm example, we conducted a *cis*-MR analysis of CRP in CHD¹ that proved the observational association is non-causal. Consistent findings were obtained by others². Subsequent GWAS of CRP have been conducted and identified variants in genes outside CRP (acting in *trans*) that associate with CRP concentration, including in genes encoding the receptor for the inflammatory cytokine IL-6, and the gene ApoC1, involved in lipid metabolism. Variants in *IL6R* that are associated with lower CRP concentration are associated with lower risk of CHD^{3,4} and variants in ApoC1 that are associated with higher concentration of CRP are associated with increased risk of CHD. Given the findings of the *cis*-MR of CRP on CHD and the known causal role of LDL-C for this outcome, from both genetic analyses and trials, using variants in APOC1 acting in *trans* to probe the causal relevance of CRP for CHD would be prone to confounding by horizontal pleiotropy. The same argument applies to the use of variants in *IL6R* for the same purpose. Signalling through the interleukin-6 receptor encoded by *IL6R* influences many inflammatory molecules beyond CRP that are the likely mediators of its effect on CHD.

Rebuttal References

1. Wensley F, Gao P, Burgess S, Kaptoge S, Di Angelantonio E, Shah T, Engert JC, Clarke R, Davey-Smith G, Nordestgaard BG, Saleheen D, Samani NJ, Sandhu M, Anand S, Pepys MB, Smeeth L, Whittaker J, Casas JP, Thompson SG, Hingorani AD, Danesh J. Association between C reactive protein and coronary heart disease: mendelian randomisation analysis based on individual participant data. *BMJ* 2011;**342**:d548.
2. Genetic Loci Associated With C-Reactive Protein Levels and Risk of Coronary Heart Disease | Genetics and Genomics | JAMA | JAMA Network. <https://jamanetwork.com/journals/jama/fullarticle/184182> (9 March 2020)
3. Sarwar N, Butterworth AS. Interleukin-6 receptor pathways in coronary heart disease: A collaborative meta-analysis of 82 studies. *The Lancet* 2012;**379**:1205–1213.
4. Swerdlow DI, Hingorani AD, Casas JP, Consortium IMR. The interleukin-6 receptor as a target for prevention of coronary heart disease: a mendelian randomisation analysis. *Lancet* 2012;**379**:1214–1224.

To ensure readers have access to these concepts, we have included additional text and the panels in a new section of the Appendix entitled: 'Reducing horizontal pleiotropy in MR analysis based on protein exposures and *cis* instruments.'). We refer to this new section on p.5 of the revised paper as follows:

"Thus, from an MR perspective, proteins are in a privileged position compared to other categories of risk factor and the use of *cis*-MR represents an optimal approach to instrument their causal effect for disease (See Appendix pages 3-5 and Appendix Figure 2-3)."

We have also added the following discussion on page 27-28.

“MR of protein exposure have been conducted before, sometimes selecting both *cis* and *trans* and sometimes even *trans* only variants. However, the use *cis* instruments for MR analysis of proteins is less prone to violation of the horizontal pleiotropy assumption than the use of *trans* instruments (see Appendix pages 3-5) and is amply illustrated by the example of CRP. Circulating CRP concentration is associated with CHD risk in non-genetic observational studies, but it was previously unclear if this association is causal, confounded or arises due to reverse causation. In 2011, in a paradigm example, we conducted a *cis*-MR analysis of CRP in CHD⁶² that proved the observational association is non-causal. Consistent findings were obtained by others⁶³. Subsequent GWAS of CRP have been conducted⁶⁴ and identified variants in genes outside CRP (acting in *trans*) that associate with CRP concentration, including in genes encoding the receptor for the inflammatory cytokine IL-6 the ApoC1, involved in lipid metabolism. Variants in both *IL6R* that are associated with lower CRP concentration are associated with lower risk of CHD^{7,28} and variants in *APOC1* that are associated with higher concentration of CRP are associated with increased risk of CHD⁶⁵. However, it would be erroneous to suppose that a *trans*-MR analysis of CRP instrumented using *APOC1* variants provides evidence of a causal role of CRP in CHD⁶⁵ since the same variants are also associated with LDL-C. Given the findings of the *cis*-MR of CRP on CHD and the known causal role of LDL-C for this outcome, from both genetic analyses and trials, using variants in *APOC1* acting in *trans* to probe the causal relevance of CRP for CHD would be prone to confounding by horizontal pleiotropy. The same argument applies to the use of variants in *IL6R* for the same purpose. Signalling through the interleukin-6 receptor encoded by *IL6R* influences many inflammatory molecules beyond CRP that are the likely mediators of its effect on CHD.”

Because *cis*-MR analysis greatly reduces but does not completely abolish the scope for horizontal pleiotropy, which can still arise, for example by confounding through LD, we have also added an additional approach to help identify potential horizontal pleiotropy using eQTL data. The section is on p.22 as follows:

“Using eQTLs to screen for horizontal pleiotropy

A key assumption of any drug target Mendelian randomization study is the absence of horizontal (pre-translational) pleiotropic pathway. For example, a variant associated with PCSK9 expression may also associate (e.g., through LD) with the expression of other genes. To explore this, we sequentially pruned the GTEx expression eQTL data for an association with the expression of a “non-target” gene within 1 MB of the encoding locus (based on a p-value threshold of $\{1 \times 10^{-8}, \dots, 1 \times 10^{-3}\}$; Appendix Figure 26). In general, across the 4 positive control loci, we did not see much influence of LD-based horizontal pleiotropy; and within a single tissue we did not observe much directional discordance. For each drug-target we do observe a few tissue-specific associations that only obtain significance after pruning potential pleiotropic variants to a very low p-value threshold. For example, *CETP* expression in the colon is only associated with CHD after removing variants that had a p-value $< 1 \times 10^{-3}$ with neighbouring genes. Note that the type of eQTL horizontal pleiotropy screening proposed here can be applied irrespective of the intended exposure, for example it could also be used in drug target MR analyses using pQTL exposures. “

19) If the MR analysis is aiming at estimating the effect of protein (P) on disease (D), the distinction between direct (ϕ G) and indirect (μ θ) effects of the protein on disease is irrelevant. The authors simply expand the P→D path to include X, which is ignored in the protein-disease MR analysis described in lines 181-182 (since they eventually only provided

an estimator for ω = the sum of direct and indirect effects of protein on disease). The real horizontal pleiotropy that they should worry about is the existence of ϕG , which could exist if 1) G has cis-effects on other proteins encoded in the same region, 2) G has trans-effects on other proteins, 3) G has effects on disease through other biological pathways that does not involve protein level change. Their lumping approach does address this. Also, this paper does not contribute to assess how MR could help to determine that ϕG could be in fact 0 (one of the key challenges in drug target validation).

Response: We fully agree that the critical horizontal pleiotropy assumption is that $\phi G = 0$. The derivations aim to convey exactly this message, and we are pleased that the reviewer came to the same conclusion.

Horizontal pleiotropy of the type $\phi G \neq 0$, can occur when a gene encodes multiple proteins (all affecting disease), or when variants within the gene are associated with other loci (and these loci cause disease). The potential influence of this later type of horizontal pleiotropy can be identified by screening the variants within a *cis* region for associations with expression of other genes. This was implemented and described on pages 22:

“Using eQTLs to screen for horizontal pleiotropy

A key assumption of any drug target Mendelian randomization study is the absence of horizontal (pre-translational) pleiotropic pathway. For example, a variant associated with PCSK9 expression may also associate (e.g., through LD) with the expression of other genes. To explore this, we sequentially pruned the GTEx expression eQTL data for an association with the expression of a “non-target” gene within 1 MB of the encoding locus (based on a p-value threshold of $\{1 \times 10^{-8}, \dots, 1 \times 10^{-3}\}$; Appendix Figure 26). In general, across the 4 positive control loci, we did not see much influence of LD-based horizontal pleiotropy; and within a single tissue we did not observe much directional discordance. For each drug-target we do observe a few tissue-specific associations that only obtain significance after pruning potential pleiotropic variants to a very low p-value threshold. For example, *CETP* expression in the colon is only associated with CHD after removing variants that had a p-value $< 1 \times 10^{-3}$ with neighbouring genes. Note that the type of eQTL horizontal pleiotropy screening proposed here can be applied irrespective of the intended exposure, for example it could also be used in drug target MR analyses using pQTL exposures. “

And pages 25:

“We further introduce an exploratory analysis to determine the influence of horizontal pleiotropy, pruning variants that associate with other genes around a locus encoding a drug target. “

20) The authors discuss at length a smaller set of <4,500 “druggable genes”, but such dimension reduction from the full protein-coding genome is not being leveraged, e.g. for analyzing more than just the selected loci, or for multiple testing correction (the authors admit their current significance cut-offs are fairly lenient).

Response: We agree that a druggable genome wide association analysis focusing on variants in and around the 4500 druggable genes could be done and would reduce the multiple testing burden compared to a classical GWAS. Indeed in our previous *Science*

Translational Medicine paper (Finan, Gaulton et al. The druggable genome and support for target identification and validation in drug development) updated the estimate of the druggable genome and designed and co-developed with illumina the DrugDev genotyping array with enhanced coverage of the genes encoding druggable targets. In that paper we wrote:

“Fulfilling the potential of GWAS (and studies using disease-focused genotyping arrays) for drug development requires mapping disease- or biomarker-associated SNPs to genes encoding druggable proteins and to their cognate drugs and drug-like compounds....

...We therefore updated the set of genes comprising the druggable genome. We then linked GWAS findings curated by the National Human Genome Research Institute and European Molecular Biology Laboratory, European Bioinformatics Institute (EMBL-EBI) GWAS catalog to this updated gene set, as well as to encoded proteins and associated drugs or drug-like compounds curated in the ChEMBL and First Databank (FDB) databases....

“In addition, to better support future genetic studies for disease-specific drug target identification and validation, we assembled the marker content of a new genotyping array designed for high-density coverage of the druggable genome and compared this focused array with genotyping arrays previously used in GWAS.”

Moreover, in our 2017 bioRxiv pre-print and 2019 *Scientific Reports* paper (Hingorani AD et al. Improving the odds of drug development success using human genomics. Modelling study) we refer to the potential of ‘druggable genome wide association studies’ and showed a figure related to that particular study design (Figure 7 of that paper).

Moreover, we already provide a discussion of the appropriate significance threshold in the Discussion. For absolute clarity we have amended this as follows:

“In other settings, for example gene-based MR analysis of all 4500 druggable genes, appropriate control of false discovery rates is clearly essential. It could be argued that applying a genome-wide association p-value threshold (e.g. 5×10^{-8}) will be needlessly conservative. Instead of applying the typical GWAS multiplicity correction one could control for the number of druggable proteins (about 5000; resulting in a 1×10^{-5} threshold).”

Also, that “cis-MR for drug target validation requires the selection of genes for druggable proteins” (line 241) is plainly wrong: others have applied MR to targets that would not fall into this category (see e.g. Sun et al., Ref. 8), and the concept of what constitutes a good drug target has changed substantially over the recent years (see e.g., Plenge et al., Nat Rev Drug Disc 2013; PMID: 23868113 for discussing a more genetics perspective).

Response: Of course, we agree that *cis*-MR can be applied to any protein regardless of druggability, provided there are appropriate genetic instruments, and nowhere in the paper do we argue that it shouldn't. However, *if the aim is drug development*, it is clear that it would be sensible to prioritise druggable proteins. We also agree that the range of druggable proteins is not fixed. Indeed, our own *Science Translational Medicine* paper expanded the druggable genome from around 2000 to over 4000 proteins. Moreover, new therapeutic

modalities e.g. RNA silencing, are extending the range of therapeutic targets beyond those that are currently amenable to the action of small molecules, peptides and monoclonal antibody therapeutics that target proteins, and which remain the mainstay of drug development.

We have therefore added the following on p.31:

“We also emphasise that the range of druggable proteins is not fixed. Indeed, our own previous paper expanded the druggable genome from around 2000 to over 4000 proteins²⁷. Moreover, new therapeutic modalities e.g. RNA silencing, are extending the range of therapeutic targets from those that are currently amenable to the action of small molecules, peptides and monoclonal antibody therapeutics that target proteins, and which remain the mainstay of drug development.”

The term “drug target MR” introduced here is thus certainly overstated: the method presented here might serve as one additional useful approach to validate drug targets through MR, but would most likely be used as one of several methods. I speculate that for most targets it will probably not turn out as the best one, but it’d be great to be convinced otherwise.

Response: The concept of drug target MR is introduced to differentiate the validation of a drug target through MR from the more traditional MR that is often focussed assessing the casual relationship of more distal exposures such as BMI (see extensive response to Reviewer 1 on this issue). The reviewer is correct that *cis*-MR or even *trans*-MR of protein exposures have been applied irrespective of the druggability of the protein; as referred to in Sun et al. However, such analysis can only provide actionable insight for drug development when the protein itself may possibly be drugged (whether with current methodologies or after future refinements). As such drug target MR can be seen as a subcategory of protein exposure MR, explicitly aiming to inform drug development. Clearly what is, and is not, druggable is continually changing, as such with the current paper we set out to introduce a general framework that can be used irrespective of type of drug target.

We also fully agree that drug target MR is only one specific form of drug target validation and envision that future challenges revolve around combining multiple (genetic as well as non-genetic) sources of evidence on the likely in human efficacy of a drug target.

The following was added, on p.31

“In this current manuscript we have exclusively focussed on Mendelian randomization as a tool for drug target validation, however many complimentary methods exist, often utilizing non-genetic cell, tissue and animal experiments. A key challenge to further improve (early) drug development will be to incorporate these differs sources of evidence to accurately predict in-human efficacy.”

and the following on p.27-28

“MR of protein exposure have been conducted before, sometimes selecting both *cis* and *trans* and sometimes even *trans* only variants. However, the use *cis* instruments for MR analysis of proteins is less prone to violation of the horizontal pleiotropy assumption than the use of *trans* instruments (see

Appendix pages 3-5) and is amply illustrated by the example of CRP. Circulating CRP concentration is associated with CHD risk in non-genetic observational studies, but it was previously unclear if this association is causal, confounded or arises due to reverse causation. In 2011, in a paradigm example, we conducted a cis-MR analysis of CRP in CHD⁶² that proved the observational association is non-causal. Consistent findings were obtained by others⁶³. Subsequent GWAS of CRP have been conducted⁶⁴ and identified variants in genes outside CRP (acting in trans) that associate with CRP concentration, including in genes encoding the receptor for the inflammatory cytokine IL-6 the ApoC1, involved in lipid metabolism. Variants in both IL6R that are associated with lower CRP concentration are associated with lower risk of CHD^{7,28} and variants in APOC1 that are associated with higher concentration of CRP are associated with increased risk of CHD⁶⁵. However, it would be erroneous to suppose that a trans-MR analysis of CRP instrumented using APOC1 variants provides evidence of a causal role of CRP in CHD⁶⁵ since the same variants are also associated with LDL-C. Given the findings of the cis-MR of CRP on CHD and the known causal role of LDL-C for this outcome, from both genetic analyses and trials, using variants in APOC1 acting in trans to probe the causal relevance of CRP for CHD would be prone to confounding by horizontal pleiotropy. The same argument applies to the use of variants in IL6R for the same purpose. Signalling through the interleukin-6 receptor encoded by IL6R influences many inflammatory molecules beyond CRP that are the likely mediators of its effect on CHD.”

REVIEWERS' COMMENTS:

Reviewer #1 (Remarks to the Author):

I appreciate the changes the authors have made to the study and the paper. I think it is a much stronger contribution now.

For what it's worth, I agree with the authors' response to reviewer 2's concern about horizontal pleiotropy from cis-pQTLs. I think evidence from GTEx and Sun et al shows that cis-eQTLs are more likely to impact multiple genes than cis-pQTLs.

There are still numerous small typos scattered through-out the paper. The manuscript could use a final thorough editorial review but otherwise I fully endorse its publication.

Reviewer #2 (Remarks to the Author):

In their revised manuscript, the authors aim to provide clarifications to my concerns by adding additional content. Among others, they extend the scope of their work by applying cis-MR to five additional drug targets using pQTLs from the INTERVAL study and conducting phenome scans across 35 traits. I continue to find merit in the manuscript for introducing a mathematical concept why cis-MR on proteins can support drug target validation and for well illustrating the various challenges and pitfalls of locus-specific MR. However, I remain unconvinced that their approach is novel, and that joint analyses of multiple unselected variants per locus will yield more than just incremental insights for the vast majority of proteins when analyzed systematically. As is, the manuscript to me does not provide sufficient evidence for several of the claims made and overall continues to lean towards overstatements, possibly best reflected in a title that is very generic instead of specifically stating what's been done and what's new. For instance,

15) Locus-specific MR to validate distinct drug targets is not a new concept, but broadly applied in industry and academia. This is evidenced by an increasing number of publications, which (on a coarse look-up) include PMID: 25726324 for IL1RN, PMID: 31558144 for F11, PMID: 31253830 for IL18, PMID: 30865797 for ACLY, or PMID: 29875488 for IL6R, IL12/23, or GP1BA among others.

16) I appreciate the expansion of the manuscript and application of cis-MR to 5 additional drug targets, although several of the selected genes have already been analyzed nearly identically with pQTL-based MR phenome-scans (across multiple GWAS traits) in the source paper (Sun et al., PMID: 29875488; Supplemental Table 16). What would have been interesting to compare in this current manuscript is: Does clumping of cis-instruments across a gene region as proposed in the manuscript yield benefits to single SNP-based cis-MR (as applied by Sun et al.) and colocalization analysis (state of the art now) when applied systematically to a larger number of drug target genes. Based on the insights presented on the four drug targets of the original manuscript this does not seem to be the case. For the 5 new genes a comparative analysis to other MR approaches would have yielded more insights for supporting the core hypothesis of the paper than a phenome scan.

18+19) I acknowledge the considerable strengths of pQTLs as instruments for cis-MR, yet the linearity is not always as simple as stated in the manuscript and rebuttle to justify that pQTLs would minimize the risk for horizontal pleiotropy. For instance, increasing numbers of cis-pQTLs are also be trans-pQTLs. Sun et al. e.g. discussed examples like MST1, PR3/SERPINA1, or GDF8/11/ACV2RB where the same instrument modulates multiple proteins of the same pathway or protein complex, limiting conclusions for the cis-drug target in isolation. Pruning for such variants (as described in 19) may confound conclusions on the biology of a target and provide a false impression on the directionality to modulate a target by drugs. Also, the authors still do not provide evidence that jointly using multiple, little characterized instruments within a 2.5kb window

not just blows up the error rate instead of making cis-MR more sensitive and reliable.

20) I am still uncertain why the druggable genome continues to be so highlighted in this manuscript - clearly, as the authors agree, pQTL-based cis-MR is universally applicable to any instrumented protein-coding gene and with new modalities such as biologics, siRNAs/ASOs, gene editing or gene therapy having become centerstage, drug development is moving away more and more from a defining a distinct subset of the genome as "druggable". As such, I do not see a need to introduce a flashy term such as "drug target MR", especially since the authors seem to agree it is not more than "protein exposure MR", just with the aim to inform drug development.

Overall, the manuscript has become very long and would benefit from streamlining.

Reviewer 1

1) I appreciate the changes the authors have made to the study and the paper. I think it is a much stronger contribution now.

For what it's worth, I agree with the authors' response to reviewer 2's concern about horizontal pleiotropy from cis-pQTLs. I think evidence from GTEx and Sun et al shows that cis-eQTLs are more likely to impact multiple genes than cis-pQTLs.

There are still numerous small typos scattered through-out the paper. The manuscript could use a final thorough editorial review but otherwise I fully endorse its publication.

Response: We thank the reviewer. We have proofread the manuscript and corrected errors.

Reviewer 2

2) In their revised manuscript, the authors aim to provide clarifications to my concerns by adding additional content. Among others, they extend the scope of their work by applying cis-MR to five additional drug targets using pQTLs from the INTERVAL study and conducting phenome scans across 35 traits. I continue to find merit in the manuscript for introducing a mathematical concept why cis-MR on proteins can support drug target validation and for well illustrating the various challenges and pitfalls of locus-specific MR. However, I remain unconvinced that their approach is novel, and that joint analyses of multiple unselected variants per locus will yield more than just incremental insights for the vast majority of proteins when analyzed systematically. As is, the manuscript to me does not provide sufficient evidence for several of the claims made and overall continues to lean towards overstatements, possibly best reflected in a title that is very generic instead of specifically stating what's been done and what's new. For instance,

Response: While we would like to acknowledge and address the reviewer's concern about insufficient evidence for claims and any overstatements, the lack of specific examples makes it challenge to provide a response. If there are specific sentences or elements that are problematic, we would be very happy to review these.

Regarding the objection to the title, our manuscript attempts to formally define drug target validation using Mendelian randomization methods and provide modelling recommendations based on theoretical considerations, which are then tested empirically. As such we believe our title "Genetic drug target validation using Mendelian randomization", properly covers the content.

3) Locus-specific MR to validate distinct drug targets is not a new concept, but broadly applied in industry and academia. This is evidenced by an increasing number of publications, which (on a coarse look-up) include PMID: 25726324 for IL1RN, PMID: 31558144 for F11, PMID: 31253830 for IL18, PMID: 30865797 for ACLY, or PMID: 29875488 for IL6R, IL12/23, or GP1BA among others.

Response: do not claim that the use of cis-variants for drug target validation is a novel approach. Indeed, we cite several prior examples (many of which we conducted ourselves). For example, to our knowledge, the first specific use of this approach to address a drug development question was in a 2010 paper from our group (Sofat, Hingorani et al., *Circulation* 2010; 121, 52-62), where "we used common genetic polymorphisms in the *CETP* gene to distinguish whether the hypertensive action of torcetrapib was mechanism based or off target, because a genetic study of these variants can be considered to be a type of natural randomized trial of a "clean" low-dose CETP inhibitor with no off-target actions.". We went on to write in that paper: "Genetic studies could be used in drug-development programs as a new source of randomized evidence for drug-target validation in humans."

Instead our paper attempts to more formally describe the merits, pitfalls, and statistical interpretation of such analyses for drug target validation, and in particular the decisions that need to be made on the selection of the many genetic instruments that might be used for

such analyses in the post-GWAS era, as well as, the emerging opportunity to use integrate proteomics data in such analyses, which we believe to be an important advance.

4) I appreciate the expansion of the manuscript and application of cis-MR to 5 additional drug targets, although several of the selected genes have already been analyzed nearly identically with pQTL-based MR phenome-scans (across multiple GWAS traits) in the source paper (Sun et al., PMID: 29875488; Supplemental Table 16). What would have been interesting to compare in this current manuscript is: Does clumping of cis-instruments across a gene region as proposed in the manuscript yield benefits to single SNP-based cis-MR (as applied by Sun et al.) and colocalization analysis (state of the art now) when applied systematically to a larger number of drug target genes. Based on the insights presented on the four drug targets of the original manuscript this does not seem to be the case. For the 5 new genes a comparative analysis to other MR approaches would have yielded more insights for supporting the core hypothesis of the paper than a phenome scan.

Response: As we mentioned in the previous rebuttal letter, as well as in the main text (page 37-38) that our proposed instrument selection strategy is agnostic of the type of MR method (as long as an account is made for LD). As such a comparison between MR estimators (while very interesting) is not pertinent to the current manuscript. We do apologize if our previous explanation did not clarify this.

In relation to the analysis of five additional drug targets that, some of which were also reported before by Sun et al., we note that our intention was merely to showcase the generalizability of our selection strategy beyond lipid targets, not to necessarily identify novel causal effect of protein perturbation. Second, we wished to display the scalability of our approach, which the reviewer had helped to highlight. Supplemental table 16 (from Sun et al) indeed lists three of the five drug targets evaluated by us, (specifically Factor 10, IL12, and IL1R1). Together with appendix table 14, however, these tables simply list the p-values for a lookup of the sentinel variant at each of these loci with outcomes from prior GWAS (not the effect size, nor the direction of effect, nor the effect on the outcome in relation to the effect in the encoded protein). Instead of focussing simply on a lookup of genetic effects, we used information on genetic effect on the concentration of the encoded protein to estimate the direction and rank order of effect of *protein* perturbation on disease outcomes, using the proposed modelling strategy delineated in the manuscript and extended this to consider 35 therapeutically relevant outcomes.

5) I acknowledge the considerable strengths of pQTLs as instruments for cis-MR, yet the linearity is not always as simple as stated in the manuscript and rebuttle to justify that pQTLs would minimize the risk for horizontal pleiotropy. For instance, increasing numbers of cis-pQTLs are also be trans-pQTLs. Sun et al. e.g. discussed examples like MST1, PR3/SERPINA1, or GDF8/11/ACV2RB where the same instrument modulates multiple proteins of the same pathway or protein complex, limiting conclusions for the cis-drug target in isolation. Pruning for such variants (as described in 19) may confound conclusions on the biology of a target and provide a false impression on the directionality to modulate a target

by drugs. Also, the authors still do not provide evidence that jointly using multiple, little characterized instruments within a 2.5kb window not just blows up the error rate instead of making cis-MR more sensitive and reliable.

Response: We are glad the reviewer now acknowledges the considerable strengths of *cis*-MR. We agree that cis-MR analysis does not preclude horizontal pleiotropy, merely that cis-MR instrumenting genetic effects through the encoded protein very substantially diminishes the chances of horizontal pleiotropy compared to other types of MR analysis. We describe using the traditional outlier and leverage statistics to detect its presence and introduce an eQTL based screening step (see methods) that further diminishes the risk of horizontal pleiotropy. With respect to the observation raised again by the reviewer that cis-pQTLs may be trans pQTLs for other proteins, we argued at length in the prior rebuttal as to why this is to be expected but why also it does not affect our argument. We introduced supplementary Figure 8 and a discussion on cis instruments on pp41-42 of the appendix which fully rebuts the point the reviewer once again refers to here. For this reason, we do not expand again on our arguments here.

The 2.5kb flanking region is merely used as an opportunistic window to decrease the potential of LD related horizontal pleiotropy. Nowhere do we suggest that this should *always* be used, and we evaluate multiple regions alternative region sizes as discussed in Supplemental Discussion section. As mentioned before (see Page 40 and references 45 and 49), the type 1 error rate is safeguarded through LD-modelling using an external reference populations.

6) I am still uncertain why the druggable genome continues to be so highlighted in this manuscript - clearly, as the authors agree, pQTL-based cis-MR is universally applicable to any instrumented protein-coding gene and with new modalities such as biologics, siRNAs/ASOs, gene editing or gene therapy having become centerstage, drug development is moving away more and more from a defining a distinct subset of the genome as “druggable”. As such, I do not see a need to introduce a flashy term such as “drug target MR”, especially since the authors seem to agree it is not more than “protein exposure MR”, just with the aim to inform drug development.

Response: We believe that our paper already adequately addresses the issue that cis-MR of proteins is relevant to understanding the disease relationships of any encoded protein but that there may be a specific therapeutic interest in a protein if it is also likely to be druggable by small molecule or monoclonal antibody therapeutics. We fully agree that siRNAs and ASOs provide alternative therapeutic strategies if an encoded protein is not readily druggable and if the expression of the gene of interest is in a tissue (e.g. liver that is accessible to siRNAs/ASOs). Indeed the approach we describe can also be used to assess the validity of targets for siRNAs/ASOs. However, proteins remain the main class of therapeutic targets and are likely to remain so for the foreseeable future.

We additionally note that the concept of drug target MR (as well as its name) has been introduced before.

7) Overall, the manuscript has become very long and would benefit from streamlining.

Response: We agree with the reviewer and have reduce the manuscript down to approximately 5000 words. We hope it is now more concise and easier to read.